# DeCoDe: Decoupling Binding Position and Molecular Conformation in 3D Ligand Diffusion for Structure-Based Drug Design

**Julong Yang**[1]   **Wen Huang**[1]   **Junhui Chen**[1]   **Jian Peng**[1]

## Abstract

Recent advances in diffusion models show promise for Structure-Based Drug Design (SBDD), which aims to generate 3D ligand molecules that bind tightly to specific protein targets. This involves jointly optimizing the ligand's 3D conformation and its binding position within the protein pocket. However, existing diffusion-based SBDD methods diffuse conformation and binding position synchronously within a high-dimensional joint space, leading to inefficient exploration and suboptimal generation quality in both aspects. To address this, we propose **DeCoDe**, a novel diffusion framework that **decouples** the diffusion processes of the binding position and molecular conformation. Our key insight is to prioritize the perturbation of the ligand's internal conformation in the early stages of the forward (noising) process, while accelerating the perturbation of its global binding position later. This design guides the reverse (denoising) process to *first coarsely position* the ligand within the pocket before *refining its detailed structure*, mimicking a more efficient, step-wise generation strategy. Extensive experiments on the CrossDocked2020 benchmark show that DeCoDe achieves significantly higher structural fidelity (with an average improvement of 18%), while maintaining competitive binding affinity and overall molecular properties compared to state-of-the-art baselines.

## 1. Introduction

Structure-Based Drug Design (SBDD) aims to discover ligand molecules with high binding affinity for specific protein targets(Anderson, 2003), enabling precise regulation of bi-

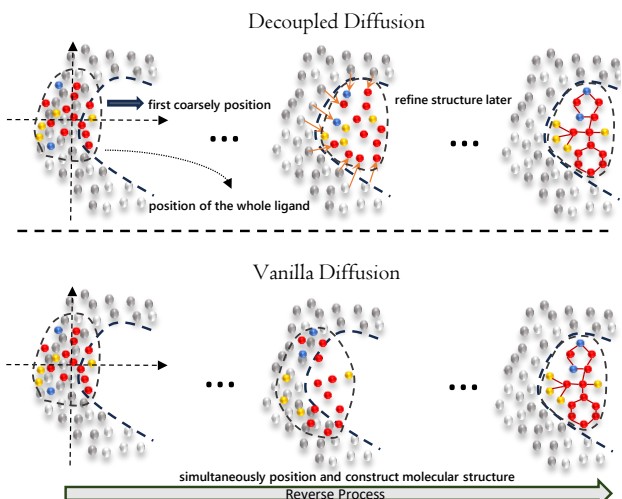

*Figure 1.* The difference between decoupled diffusion and vanilla diffusion. The vanilla diffusion jointly denoise both the molecular structure and binding position of ligands within a single entangled process, leading to unnecessary complexity into the optimization; While our decoupled diffusion employs divide-and-conquer strategy: first coarse positioning the ligand within the pocket constrains the search space for the subsequent molecular structure design.

ological functions for therapeutic applications. Despite its transformative potential in drug discovery, SBDD faces significant challenges due to the combinatorial explosion of feasible chemical space(Ragoza et al., 2022) and the need for atomic-level geometric precision.

Generative AI models(Li et al., 2021; Ragoza et al., 2022; Powers et al., 2022; ZHANG et al., 2023) have emerged as promising tools for designing 3D ligand structures conditioned on protein pockets. Early approaches(Luo et al., 2021; Liu et al., 2022; Peng et al., 2022) leverage autoregressive atom-by-atom generation, modeling protein-ligand interactions via neural networks. Recent diffusion models(Guan et al., 2023a; Schneuing et al., 2023; Lin et al., 2022; Huang et al., 2024a;b;c; Gu et al., 2024; Gao et al., 2024) have shown state-of-the-art performance by denoising ligand coordinates and atom types jointly using SE(3)-equivariant networks(Hoogeboom et al., 2022).

However, these diffusion methods largely neglect a key

[1]College of Computer Science, Sichuan University. Correspondence to: Jian Peng <jianpeng@scu.edu.cn>, Wen Huang <562421007@qq.com>.

*Proceedings of the $43^{rd}$ International Conference on Machine Learning*, Seoul, South Korea. PMLR 306, 2026. Copyright 2026 by the author(s).

biophysical insight: The ligand is a pre-formed molecular entity that exists independently prior to its binding with the pocket. Consequently, its atoms' coordinates in the protein-ligand complex are governed by two distinct factors: binding position (global placement relative to the pocket) and intrinsic molecular conformation (internal geometry). Existing SBDD diffusion models perturb and reconstruct ligand atoms' coordinates in a fully coupled manner. This forces models to simultaneously solve two interdependent sub-problems: ①Identifying plausible binding locations within the pocket(Global positioning), ②Generating chemically valid 3D molecular geometries(Structural design). The high-dimensional joint solution space (position × conformation) leads to inefficient exploration and suboptimal generation quality, as the model struggles to coordinate both objectives at once. While DecompDiff(Guan et al., 2023b) partially addresses this issue by generating molecular substructures within pre-selected pocket regions—its primary objective is molecular decomposition rather than disentangling binding position and internal conformation. Moreover, its region selection relies on external tools, limiting end-to-end learnability and generalizability.

We argue that decoupling these subproblems simplifies the generative process by ❶*Coarse positioning first*: Roughly locating the ligand within the pocket constrains the search space for molecular structure design; ❷*Structure refinement later*: Generating structure details within a localized region is more tractable than global search. This divide-and-conquer strategy aligns with human intuition in drug design (e.g., first sketching binding motifs, then elaborating functional groups).

Inspired by this, we propose DeCoDe (**De**coupled Binding Position and Molecular **Co**nformation **De**sign), a novel diffusion framework that explicitly disentangles the generation of binding position and molecular conformation of ligands. Specifically, we implement this via **Diffusion decoupling** and **Time-dependent diffusion scheduling**. The former decouples the ligand coordinates into binding position(ligand centroid coordinates relative to pocket centroid) and molecular conformation(atom coordinates relative to ligand centroid) for respective diffusion. The latter adopts asynchronous diffusion scheduling for them: Early steps add noise primarily to conformation; late steps accelerate position perturbation. This induces the reverse process to first coarsely position the ligand within the protein pocket before refining its detailed structure, as illustrated in Figure 1. Our contributions are summarized as:

- We propose **DeCoDe**, a novel framework to explicitly decouple the diffusion of global binding position and intrinsic molecular conformation for SBDD, enabling precise and independent modulation of the two components.

- We design a **time-dependent noise-scheduling** strategy that induces a coarse-to-fine generation hierarchy, encouraging the model to first place the ligand in a plausible binding region before refining its detailed geometry—a process that aligns better with the human intuition in drug design.

- Our method can integrate with other SBDD diffusion models, boosting their generation quality while not relying on any external support. When applied to TargetDiff and IRDiff on the CrossDocked2020 benchmark, it reduces the Jensen–Shannon divergence of the bone distance distributions by an average of 18%, achieving state-of-the-art structural fidelity while preserving competitive binding affinity.

## 2. Related Work

**Structure-Based Drug Design** aims to generate 3D ligand molecules that bind to specific protein targets (Anderson, 2003). Early approaches generate SMILES strings (Skalic et al., 2019; Xu et al., 2021a) or 2D molecular graphs (Tan et al., 2023) conditioned on protein context, but they lack the explicit 3D geometry critical for binding. Subsequent methods move to the 3D space: some represent ligands as voxelized grids (Ragoza et al., 2022; O Pinheiro et al., 2023; Pinheiro et al., 2024), while more methods tackle SBDD in the continuous point cloud position of atoms. Autoregressive models, for instance, place atoms or fragments sequentially(Luo et al., 2021; Liu et al., 2022; Peng et al., 2022; ZHANG et al., 2023; Zhang & Liu, 2023). However, these methods typically suffer from error accumulation and the problem of determining the optimal generation order. In contrast, one-shot generative approaches—notably diffusion models(Ho et al., 2020) and Bayesian Flow Networks (BFNs)(Graves et al., 2023; Qu et al., 2024)—show promising performance and greater reliability.

**Diffusion Models for SBDD** have recently emerged as the dominant paradigm. Pioneering works (Guan et al., 2023a; Lin et al., 2022; Schneuing et al., 2023) establish SE(3)-equivariant denoising frameworks that jointly diffuse atomic coordinates and types in the protein–ligand complex frame. Building on them, BINDDM (Huang et al., 2024a) improves binding affinity by mining binding-relevant sub-complexes; IRDiff (Huang et al., 2024b) and IPDiff (Huang et al., 2024c) incorporate protein-ligand interaction—either via high-affinity reference structures or a modified forward process—though both rely on pre-training blocks on other datasets. DecompDiff (Guan et al., 2023b) introduces a decomposed prior to focus on molecular substructure design, but its effectiveness depends heavily on external tools and yields limited improvement in molecular conformation. D3FG (Lin et al., 2023) also uses prior knowledge of fragment motifs to model on the molecule substructure level.

AliDiff (Gu et al., 2024) proposes an energy-guided soft-DPO (Rafailov et al., 2023; Wallace et al., 2024) method to fine-tune diffusion models, thereby steering them to prefer high-affinity ligands, while SBE-Diff (Gao et al., 2024) addresses the overlooked issue of ligand specificity by optimizing a delta-score that measures pocket selectivity.

Despite these advances, existing SBDD diffusion models typically treat ligand binding position and internal conformation as a single entangled variable, overlooking their distinct biophysical roles. Our work directly addresses this gap by explicitly decoupling the diffusion processes of binding position and molecular conformation—aligning the generative process with the underlying biophysics and enabling more efficient and realistic ligand generation.

## 3. Preliminary

For SBDD, the goal of generative modeling is to sample a ligand molecule that binds to a given protein pocket. We represent the protein and ligand as $\mathcal{P} = \{(\mathbf{x}_P^{(i)}, \mathbf{v}_P^{(i)})\}_{i=1}^{N_P}$ and $\mathcal{M} = \{(\mathbf{x}_M^{(i)}, \mathbf{v}_M^{(i)})\}_{i=1}^{N_M}$, where $N_P$ (resp. $N_M$) denote the number of atoms of the protein $\mathcal{P}$ (resp. the ligand $\mathcal{M}$). $\mathbf{x} \in \mathbb{R}^3$ is the atomic coordinate, and $\mathbf{v} \in \mathbb{R}^K$ is a one-hot vector encoding the atom type($K$ types in total). In this paper, we adopt the following notational conventions: matrices are denoted by uppercase boldface letters (e.g., $\mathbf{X}$), and $\mathbf{x}_i$ refers to the $i$-th row vector of $\mathbf{X}$. For brevity, we stack positions and types into matrices: $\mathbf{X}_M \in \mathbb{R}^{N_M \times 3}$, $\mathbf{V}_M \in \mathbb{R}^{N_M \times K}$, and similarly for the protein $\mathbf{P} = [\mathbf{X}_P, \mathbf{V}_P]$. The SBDD task is then formulated as modeling the conditional distribution $p(\mathbf{M}|\mathbf{P})$, where $\mathbf{M} = [\mathbf{X}_M, \mathbf{V}_M]$.

**DDPMs for SBDD** Denoising Diffusion Probabilistic Models (DDPMs) (Ho et al., 2020), equipped with SE(3)-equivariant architectures, have become the de facto standard for SBDD (Guan et al., 2023a; Schneuing et al., 2023; Lin et al., 2022). These models jointly diffuse atomic positions and types of the ligand, while the atom count $N_M$ is either sampled from an empirical distribution (Guan et al., 2023a) or predicted by neural networks (Lin et al., 2022); molecular bonds are typically inferred as a post-processing step.

The forward diffusion process gradually injects noise into the ligand over $T$ steps via a Markov chain. Since the protein remains fixed, we drop the subscript $M$ and denote the ligand at time $t$ as $\mathbf{M}_t = [\mathbf{X}_t, \mathbf{V}_t]$. The transition kernel is defined as:

$$q(\mathbf{M}_t|\mathbf{M}_{t-1}, \mathbf{P}) = \mathcal{N}(\mathbf{X}_t; \sqrt{1-\beta_t}\mathbf{X}_{t-1}, \beta_t \boldsymbol{I}) \cdot \\ \mathcal{C}(\mathbf{V}_t|(1-\beta_t)\mathbf{V}_{t-1} + \beta_t/K), \quad (1)$$

where $\mathcal{N}$ and $\mathcal{C}$ denote Gaussian and categorical distributions, respectively, and $\beta_t$ is a fixed noise schedule. Let

$\alpha_t = 1 - \beta_t$ and $\bar{\alpha}_t = \prod_{s=1}^{t} \alpha_s$, The marginal distribution $q(\mathbf{M}_t|\mathbf{M}_0, \mathbf{P})$ admits a closed form:

$$q(\mathbf{M}_t|\mathbf{M}_0, \mathbf{P}) = \mathcal{N}(\mathbf{X}_t; \sqrt{\bar{\alpha}_t}\mathbf{X}_0, (1-\bar{\alpha}_t)\boldsymbol{I}) \cdot \\ \mathcal{C}(\mathbf{V}_t|\bar{\alpha}_t\mathbf{V}_0 + (1-\bar{\alpha}_t)/K). \quad (2)$$

This enables efficient training by sampling arbitrary time steps $t$. The corresponding posterior also has a closed-form expression:

$$q(\mathbf{M}_{t-1}|\mathbf{M}_t, \mathbf{M}_0, \mathbf{P}) = \mathcal{N}(\mathbf{X}_{t-1}; \tilde{\boldsymbol{\mu}}(\mathbf{X}_t, \mathbf{X}_0), \tilde{\beta}_t\boldsymbol{I}) \cdot \\ \mathcal{C}(\mathbf{V}_{t-1}|\tilde{\boldsymbol{c}}(\mathbf{V}_t, \mathbf{V}_0)), \quad (3)$$

with $\tilde{\boldsymbol{\mu}}(\mathbf{X}_t, \mathbf{X}_0) = \frac{\sqrt{\bar{\alpha}_{t-1}}\beta_t}{1-\bar{\alpha}_t}\mathbf{X}_0 + \frac{\sqrt{\alpha_t}(1-\bar{\alpha}_{t-1})}{1-\bar{\alpha}_t}\mathbf{X}_t$, $\tilde{\beta}_t = \frac{1-\bar{\alpha}_{t-1}}{1-\bar{\alpha}_t}\beta_t$, and $\tilde{\boldsymbol{c}}(\mathbf{V}_t, \mathbf{V}_0) = \frac{\boldsymbol{c}^*}{\sum_{k=1}^{K} c_k^*}$ where $\boldsymbol{c}^*(\mathbf{V}_t, \mathbf{V}_0) = [\alpha_t\mathbf{V}_t + (1-\alpha_t)/K] \odot [\bar{\alpha}_{t-1}\mathbf{V}_0 + (1-\bar{\alpha}_{t-1})/K]$. In the reverse (generative) process, a neural network parameterized by $\theta$ predicts the clean data $(\hat{\mathbf{X}}_0, \hat{\mathbf{V}}_0)$ and approximates the reverse kernel as:

$$p_\theta(\mathbf{M}_{t-1}|\mathbf{M}_t, \mathbf{P}) = \mathcal{N}(\mathbf{X}_{t-1}; \tilde{\boldsymbol{\mu}}(\mathbf{X}_t, \hat{\mathbf{X}}_0), \tilde{\beta}_t\boldsymbol{I}) \cdot \\ \mathcal{C}(\mathbf{V}_{t-1}|\tilde{\boldsymbol{c}}(\mathbf{V}_t, \hat{\mathbf{V}}_0)). \quad (4)$$

## 4. Method

Most existing diffusion-based approaches for SBDD follow the general diffusion framework outlined in Section 3. However, this framework couples the global binding position and the intrinsic molecular conformation into a single diffusion process, which complicates the generative space unnecessarily. In this section, we introduce **DeCoDe** (illustrated as Figure 2), a general diffusion framework that explicitly separates the modeling of the global binding position from the internal molecular structure. DeCoDe consists of two key components: (1) *diffusion decoupling*, which formulates independent diffusion processes for position and structure, and (2) *time-dependent diffusion scheduling*, which adapts the noise schedule for each component. We detail these components in Sections 4.1 and 4.2, respectively.

### 4.1. Diffusion Decoupling

As discussed earlier, the observed atomic coordinates of a ligand $\mathbf{X}$ are governed by two distinct geometric factors: (i) the global binding position relative to the protein pocket, and (ii) the internal molecular conformation. To disentangle these, we decompose $\mathbf{X}$ using its center of mass (CoM):

$$\mathbf{x}^c = \frac{1}{N_M}\sum_{i=1}^{N_M}\mathbf{x}_i, \qquad \mathbf{X}^s = \mathbf{X} - \mathbf{1}(\mathbf{x}^c)^\top, \quad (5)$$

where $\mathbf{1} \in \mathbb{R}^{N_M}$ is an all-ones vector. Thus, $\mathbf{x}^c$ represents the binding position, and the centroid-centered coordinates $\mathbf{X}^s$ corresponds to the internal molecular structure, which is translation-invariant. Using the reparameterization trick,

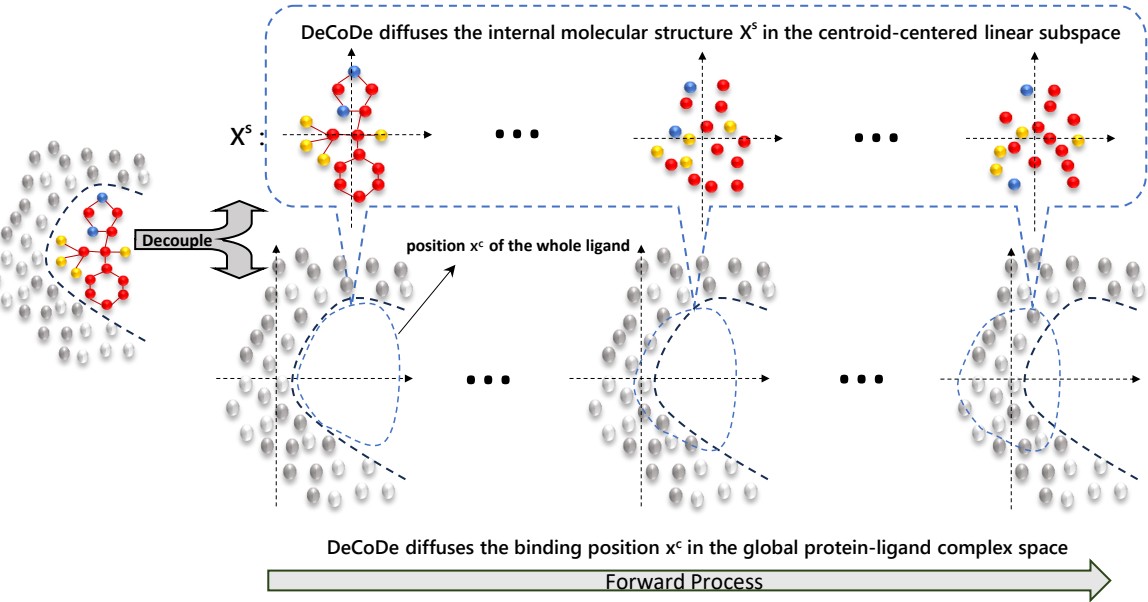

*Figure 2.* Overview of DeCoDe. Our DeCoDe decouples the binding position and the molecular structure of ligands, and diffuses them in different space: the binding position is diffused in the original protein-ligand complex reference system (the blue circle denotes the whole ligand molecule and its position is ultimately subjected to $\mathcal{N}(\mathbf{0}, \frac{1}{N_M}\boldsymbol{I}_3)$ with the forward process); while the molecular structure is diffused in the centroid-centered linear subspace. This decoupling mechanism allows for the precise modulation of the individual components in the ligand diffusion process.

we decompose the transition kernel of the forward diffusion process in the same manner:

$$
\begin{aligned}
\mathbf{X}_t =& \sqrt{1-\beta_t}\mathbf{X}_{t-1} + \sqrt{\beta_t}\epsilon_t \\
=& \sqrt{1-\beta_t}\left(\mathbf{x}_{t-1}^c + \mathbf{X}_{t-1}^s\right) + \sqrt{\beta_t}\left(\epsilon_t^c + \epsilon_t^s\right) \\
=& \sqrt{1-\beta_t}\mathbf{x}_{t-1}^c + \sqrt{\beta_t}\epsilon_t^c \qquad \Longleftrightarrow \quad \mathbf{x}_t^c \\
&+ \sqrt{1-\beta_t}\mathbf{X}_{t-1}^s + \sqrt{\beta_t}\epsilon_t^s, \qquad \Longleftrightarrow \quad \mathbf{X}_t^s
\end{aligned}
\tag{6}
$$

where $\epsilon_t \sim \mathcal{N}(\mathbf{0}, \boldsymbol{I}_{N_M \times 3})$, and $\epsilon_t^c$, $\epsilon_t^s$ are the decomposed noise terms for the position and structure, respectively. As shown in previous works (Xu et al., 2021b; Hoogeboom et al., 2022), $\epsilon_t^s$ is centroid-centered Gaussian noise, ensuring that $\mathbf{X}^s$ evolves in a translation-invariant subspace with its CoM fixed at zero. Conversely, $\epsilon_t^c$ follows $\mathcal{N}(\mathbf{0}, \frac{1}{N_M}\boldsymbol{I}_3)$ and is added solely to $\mathbf{x}^c$ thereby perturbing only the global ligand position. The marginal distributions for $\mathbf{x}_t^c$ and $\mathbf{X}_t^s$ follow directly from Equation (2) with appropriate substitutions. We can therefore independently sample $\epsilon_t^c$ and $\epsilon_t^s$ and construct the training tuple $\mathbf{M}_t = [\mathbf{X}_t^s, \mathbf{x}_t^c, \mathbf{V}_t]$ at any diffusion step $t$. Accordingly, the posterior distribution factorizes:

$$
\begin{aligned}
q(\mathbf{M}_{t-1}|\mathbf{M}_t, \mathbf{M}_0, \mathbf{P}) =& \mathcal{N}(\mathbf{X}_{t-1}^s; \tilde{\boldsymbol{\mu}}(\mathbf{X}_t^s, \mathbf{X}_0^s), \tilde{\beta}_t \boldsymbol{I}^s) \cdot \\
& \mathcal{N}(\mathbf{x}_{t-1}^c; \tilde{\boldsymbol{\mu}}(\mathbf{x}_t^c, \mathbf{x}_0^c), \tilde{\beta}_t \boldsymbol{I}^c) \cdot \\
& \mathcal{C}(\mathbf{V}_{t-1}|\tilde{\boldsymbol{c}}(\mathbf{V}_t, \mathbf{V}_0)).
\end{aligned}
\tag{7}
$$

with $\tilde{\boldsymbol{\mu}}$ and $\tilde{\boldsymbol{c}}$ defined as in Equation (3). Given a trained network $\phi_\theta$ that predicts $(\hat{\mathbf{X}}_0^s, \hat{\mathbf{x}}_0^c, \hat{\mathbf{V}}_0)$, the reverse kernel $p_\theta(\mathbf{M}_{t-1}|\mathbf{M}_t, \mathbf{P})$ is obtained by substituting these esti-

mates into Equation (7). Critically, this decoupling preserves the invariance of the likelihood $p_\theta(\mathbf{M}_0|\mathbf{P})$ as proven in Appendix A.

**Parameterization Decoupling.** Previous works(Guan et al., 2023a;b; Huang et al., 2024a;b;c) employ protein-fixed SE(3)-equivariant GNNs, where protein coordinates remain static during message passing. While this ensures the likelihood-invariance, it forces ligand atoms to learn both binding position's and internal structure's dynamics, mixing their gradients and thus increasing optimization difficulty. To match the decoupled diffusion process, we reformulate the denoising network as:

$$
\left(\hat{\mathbf{X}}_0^s, \hat{\mathbf{x}}_0^c, \hat{\mathbf{V}}_0\right) = \phi_\theta(\mathbf{M}_t, t, \mathbf{P}) = \phi_\theta([\mathbf{X}_t^s, \mathbf{x}_t^c, \mathbf{V}_t], t, \mathbf{P}). \tag{8}
$$

Full ligand coordinates are reconstructed as $\mathbf{X} = \mathbf{X}^s + \mathbf{1}(\mathbf{x}^c)^\top$ before message passing. At layer $l$, hidden states $\mathbf{H}^l$ and coordinates $\mathbf{X}^l$ are updated alternately as follows:

$$
\begin{aligned}
\mathbf{h}_i^{l+1} =& \mathbf{h}_i^l + \sum_{j \in \mathcal{N}_i} f_h^l\left(d_{ij}^l, \mathbf{h}_i^l, \mathbf{h}_j^l, \mathbf{e}_{ij}; \theta_h\right) \\
\Delta\mathbf{x}_i =& \sum_{j \in \mathcal{N}_i} \left(\mathbf{x}_i^l - \mathbf{x}_j^l\right) \cdot f_x^l\left(d_{ij}^l, \mathbf{h}_i^{l+1}, \mathbf{h}_j^{l+1}, \mathbf{e}_{ij}; \theta_x\right) \\
\overline{\Delta\mathbf{x}_{\text{protein}}} =& \frac{1}{N_P} \sum_{k \in \mathcal{P}} \Delta\mathbf{x}_k \\
\mathbf{x}_i^{l+1} =& \begin{cases} \mathbf{x}_i^l + \Delta\mathbf{x}_i, & \text{for} \quad i \in \mathcal{M} \\ \mathbf{x}_i^l + \overline{\Delta\mathbf{x}_{\text{protein}}}, & \text{for} \quad i \in \mathcal{P} \end{cases}
\end{aligned}
\tag{9}
$$

where $d_{ij} = \|\mathbf{x}_i - \mathbf{x}_j\|$, $\mathcal{N}_i$ is the set of $k$-nearest neighbors of atom $i$, and $\mathbf{e}_{ij}$ denotes edge features that indicate the connection is between protein atoms, ligand atoms or protein atom and ligand atom. The key modification (highlighted in blue) is that the entire protein is allowed to translate as a rigid body—geometrically equivalent to translating the entire ligand oppositely—thereby enabling the model to learn optimal ligand placement by shifting the protein context. This relieves ligand atoms from global positioning, letting them focus on forming proper internal structures. To preserve model's equivariance, after the last layer we translate the entire complex so that the protein's CoM returns to the origin. This operation is equivalent to applying an equal but opposite translation to the whole ligand while keeping the protein context static. The final outputs are:

$$\Delta \mathbf{x}_{\text{protein}} = \frac{1}{N_P} \sum_{k \in \mathcal{P}} \mathbf{x}_k^L$$
$$\hat{\mathbf{X}}_0 = \mathbf{X}^L - \Delta \mathbf{x}_{\text{protein}}$$
$$\hat{\mathbf{x}}_0^c = \mathbf{x}^c - \Delta \mathbf{x}_{\text{protein}} \qquad (10)$$
$$\hat{\mathbf{X}}_0^s = \hat{\mathbf{X}}_{0,\text{ligand}} - \mathbf{1}(\hat{\mathbf{x}}_0^c)^\top$$
$$\hat{\mathbf{V}}_0 = \text{SoftMax}(\text{MLP}(\mathbf{H}_{\text{ligand}}^L)),$$

where $\mathbf{X}^L$ and $\mathbf{H}^L$ denote the coordinate and embedding outputs of the final GNN layer, and $\hat{\mathbf{X}}_{0,\text{ligand}}$ and $\mathbf{H}_{\text{ligand}}^L$ are obtained by extracting the rows corresponding to ligand atoms.

This design decouples the learning of the binding position (effectively assigned to the protein's learned translation) from the learning of the internal molecular structure (handled by the ligand atoms), while strictly maintaining SE(3)-equivariance. As a result, the modeling burden is distributed more naturally, reducing the overall learning difficulty.

**Training Objective.** We adopt a training objective similar to Guan et al. (2023a). For each diffusion step $t-1$, the loss terms for molecular structure, binding position, and atom types are defined as:

$$\mathbf{L}_{t-1}^{(x^s)} = \frac{1}{2\tilde{\beta}_t^2} \sum_{i=1}^{N_M} \|\tilde{\boldsymbol{\mu}}(\mathbf{x}_{i,t}^s, \mathbf{x}_{i,0}^s) - \tilde{\boldsymbol{\mu}}(\mathbf{x}_{i,t}^s, \hat{\mathbf{x}}_0^s)\|^2$$
$$= \gamma_t \sum_{i=1}^{N_M} \|\mathbf{x}_{i,0}^s - \hat{\mathbf{x}}_{i,0}^s\|^2$$
$$\mathbf{L}_{t-1}^{(x^c)} = \gamma_t \sum_{i=1}^{N_M} \|\mathbf{x}_0^c - \hat{\mathbf{x}}_0^c\|^2 \qquad (11)$$
$$\mathbf{L}_{t-1}^{(v)} = \sum_{i=1}^{N_M} \sum_{k=1}^{K} \tilde{\boldsymbol{c}}(\mathbf{v}_{i,t}, \mathbf{v}_{i,0})_k \log \frac{\tilde{\boldsymbol{c}}(\mathbf{v}_{i,t}, \mathbf{v}_{i,0})_k}{\tilde{\boldsymbol{c}}(\mathbf{v}_{i,t}, \hat{\mathbf{v}}_{i,0})_k},$$

where $\mathbf{x}_{i,0}$ and $\mathbf{v}_{i,0}$ denote the ground-truth coordinate and atom type of the $i$-th ligand atom, and $\hat{\mathbf{x}}_{i,0}, \hat{\mathbf{v}}_{i,0}$ are the corresponding predictions from the model, the coefficient

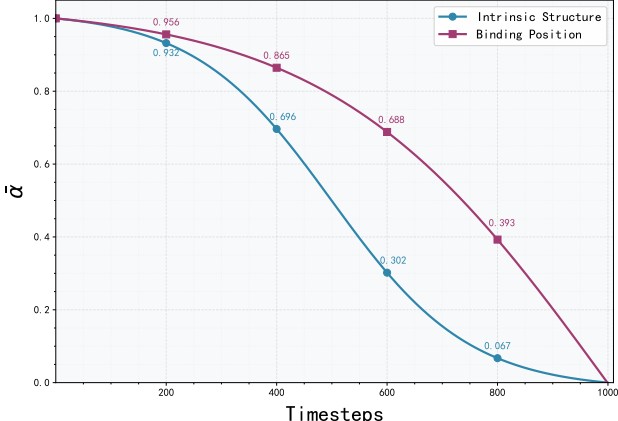

*Figure 3.* The asynchronous scheduling $\bar{\alpha}$ curves for molecular structure and binding position. Early steps primarily perturb the molecular structure, whereas later steps introduce more noise to the binding position. We implement them via the "advance schedule" proposed by Peng et al. (2023), referring Appendix D for details.

$\gamma_t = \frac{\bar{\alpha}_{t-1} \beta_t^2}{2\tilde{\beta}_t^2 (1-\bar{\alpha}_t)^2}$. Following the common practice in diffusion models (Ho et al., 2020), we set $\gamma_t = \frac{\bar{\alpha}_t}{1-\bar{\alpha}_t}$ to align with the standard simplified denoising objective (derivation in Appendix B). The total loss is a weighted sum:

$$\mathbf{L} = \mathbf{L}_{t-1}^{(x^c)} + \mathbf{L}_{t-1}^{(x^s)} + \lambda \mathbf{L}_{t-1}^{(v)}, \qquad (12)$$

with $\lambda$ a hyperparameter balancing coordinate and categorical losses. The overall training and sampling procedures are summarized in Appendix C.

### 4.2. Time-dependent Diffusion Scheduling

Decoupling the diffusion process enables independent control over the noise schedules for the binding position ($\beta_t^c$) and the molecular structure ($\beta_t^s$). Since the binding position is a low-frequency signal compared to the detailed molecular structure, we schedule the noise levels so that the binding position is roughly localized early in the reverse process, after which the structural details are refined. Concretely, we design distinct $\bar{\alpha}$ schedules for $\mathbf{x}^c$ and $\mathbf{X}^s$, as illustrated in Figure 3. During the forward diffusion, structural noise dominates in early steps, while positional noise increases more rapidly in later steps. This induces the reverse process to first coarsely position the ligand within the pocket (low-frequency), then refine its molecular structure (high-frequency). This strategy: (1) Reduces the search space for structure generation (localized search vs. global); (2) Aligns with biological intuition (binding site first, then atomic details); (3) Matches the low-to-high frequency progression inherent in diffusion models.

# 5. Experiments

## 5.1. Experimental Settings

**Datasets.** We follow prior work(Luo et al., 2021; Peng et al., 2022; Guan et al., 2023a) and train on the Cross-Docked2020 dataset (Francoeur et al., 2020). Using the same preprocessing and split as Luo et al. (2021), we retain only high-quality docking poses (RMSD < 1 Åfrom the ground truth) and ensure protein diversity (sequence identity < 30%). The final training set contains 100,000 complexes, and evaluation is performed on 100 held-out complexes that contain novel protein structures.

**Baselines.** We compare our method with six state-of-the-art diffusion-based SBDD methods, selected based on their leading performance on the CrossDocked2020 benchmark: **TargetDiff** (Guan et al., 2023a), **DecompDiff** (Guan et al., 2023b), **BINDDM** (Huang et al., 2024a), **IRDiff** (Huang et al., 2024b), **IPDiff** (Huang et al., 2024c), and **AliDiff** (Gu et al., 2024). Among them, AliDiff, which fine-tunes IPDiff with a soft-DPO objective for enhanced binding affinity, represents the current best-performing method. To isolate the contribution of our proposed DeCoDe framework, we apply it to two representative architectures, TargetDiff and IRDiff, resulting in their decoupled versions (Target)**DeCoDe** and **IRDeCo**(De), respectively.

**Implementation Details.** The DeCoDe framework does not modify the neural network parameters or architecture of the base models. For both DeCoDe and IRDeCo, we use the exact same SE(3)-equivariant graph networks as their original counterparts—the only change is allowing the whole protein pocket to translate rigidly during processing, as Equation (9). We recommend reading their respective paper for more architectural details, if interested. For the decoupled diffusion, we employ the half-advance schedule for the binding position and the advance schedule for the internal structure (details in Appendix D), with the same parameters $s_1 = 0.9999$, $s_T = 0.0001$ and $w = 4$.

**Evaluation Metrics.** We evaluate generated ligands along three dimensions: **molecular structure**, **target binding affinity**, and **molecular properties**. For **molecular structure**, we compute the Jensen–Shannon divergence (JSD) between the empirical distributions of inter-atomic and bond distances in generated vs. reference molecules—lower JSD indicates better structural realism. For **binding affinity**, we use AutoDock Vina(Eberhardt et al., 2021) to compute four standard metrics following Luo et al. (2021); Ragoza et al. (2022); Guan et al. (2023a): (1) *Vina Score*: direct affinity estimate for generated ligand molecules; (2) *Vina Min*: score after local energy minimization; (3) *Vina Dock*: score after full re-docking refinement; (4) *High Affinity*: fraction of generated ligands outperforming the native ligand in binding

*Table 1.* Jensen-Shannon divergence between bond distance distributions of the ground-truth molecules and the generated molecules, and lower values indicate better performances. "-", "=", and ":" represent single, double, and aromatic bonds, respectively. We highlight the best two results with **bold text** and underlined text, respectively. DeCoDe is based on TargetDiff, and IRDeCo(De) corresponds to IRDiff.

| Bond | DeCoDe | Target Diff | Decomp Diff | BINDDM | IPDiff | IRDiff | IRDeCo |
|------|--------|-------------|-------------|--------|--------|--------|--------|
| C−C | 0.321 | 0.369 | 0.359 | 0.380 | 0.386 | 0.439 | **0.305** |
| C=C | 0.510 | 0.505 | 0.537 | **0.229** | 0.245 | 0.272 | 0.251 |
| C−N | 0.332 | 0.363 | 0.344 | 0.265 | 0.298 | 0.302 | **0.258** |
| C=N | 0.531 | 0.550 | 0.584 | 0.245 | 0.238 | 0.255 | **0.230** |
| C−O | 0.348 | 0.421 | 0.376 | 0.329 | 0.366 | 0.371 | **0.303** |
| C=O | 0.369 | 0.461 | 0.374 | 0.249 | 0.353 | 0.361 | **0.220** |
| C:C | 0.192 | 0.263 | 0.251 | 0.282 | 0.169 | 0.214 | **0.161** |
| C:N | 0.210 | 0.235 | 0.269 | 0.130 | 0.128 | 0.209 | **0.117** |

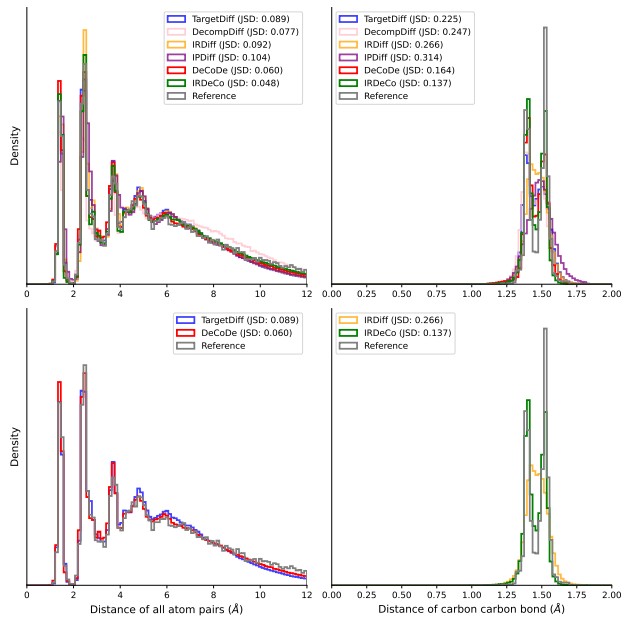

*Figure 4.* Comparing the distribution for distances of all-atom and carbon-carbon pairs for reference molecules in the test set and model-generated molecules. Jensen-Shannon divergence (JSD) between two distributions is reported.

score per target. For **molecular properties**, we report: (1) *QED* (Quantitative Estimate of Drug-likeness); (2) *SA* (Synthetic Accessibility); (3) *Diversity*, defined as the average pairwise Tanimoto dissimilarity across generated ligands (Luo et al., 2021). All sampling protocols, post-processing (e.g., bond inference), and evaluation procedures strictly follow Guan et al. (2023a) to ensure a fair comparison.

## 5.2. Main Results

**Molecular Structures.** As shown in Figure 4, our models, especially IRDeCo, best match the reference distributions of all-atom and carbon-carbon distances, achieving the lowest JSDs (0.137 for C–C bonds; 0.048 for all-atom pairs). This indicates a superior ability to capture realistic atomic

*Table 2.* Summary of different properties of reference molecules and molecules generated by our model and other baselines. (↑) / (↓) denotes a larger / smaller number is better. Top 2 results are highlighted with **bold text** and underlined text, respectively. DeCoDe is based on TargetDiff, and IRDeCo(De) corresponds to IRDiff.

| Methods | Vina Score (↓) | | Vina Min (↓) | | Vina Dock (↓) | | High Affinity (↑) | | QED (↑) | | SA (↑) | | Diversity (↑) | |
| --- | --- | --- | --- | --- | --- | --- | --- | --- | --- | --- | --- | --- | --- | --- |
| | Avg. | Med. | Avg. | Med. | Avg. | Med. | Avg. | Med. | Avg. | Med. | Avg. | Med. | Avg. | Med. |
| Reference | -6.36 | -6.46 | -6.71 | -6.49 | -7.45 | -7.26 | - | - | 0.48 | 0.47 | 0.73 | 0.74 | - | - |
| DecompDiff | -5.67 | -6.04 | -7.04 | -7.09 | -8.39 | -8.43 | 64.4% | 71.0% | 0.45 | 0.43 | 0.61 | 0.60 | 0.68 | 0.68 |
| BINDDM | -5.92 | -6.81 | -7.29 | -7.34 | -8.41 | -8.37 | 64.8% | 71.6% | 0.51 | 0.52 | 0.58 | 0.58 | **0.75** | **0.74** |
| IPDiff | -6.42 | -7.01 | -7.45 | -7.48 | -8.57 | -8.51 | 69.5% | 75.5% | 0.52 | 0.53 | 0.61 | 0.59 | 0.74 | 0.73 |
| AliDiff | **-7.07** | **-7.95** | **-8.09** | **-8.17** | **-8.90** | **-8.81** | **73.4%** | **81.4%** | 0.50 | 0.50 | 0.57 | 0.56 | 0.73 | 0.71 |
| TargetDiff | -5.47 | -6.30 | -6.64 | -6.83 | -7.80 | -7.91 | 58.1% | 59.1% | 0.48 | 0.48 | 0.58 | 0.58 | 0.72 | 0.71 |
| DeCoDe | -6.19 | -6.45 | -6.81 | -6.78 | -7.79 | -7.81 | 56.9% | 54.7% | 0.48 | 0.48 | 0.60 | 0.59 | 0.71 | 0.71 |
| IRDiff | -6.03 | -6.89 | -7.27 | -7.37 | -8.42 | -8.42 | 67.4% | 72.7% | **0.53** | **0.54** | 0.59 | 0.58 | 0.72 | 0.72 |
| IRDeCo | -6.69 | -7.08 | -7.49 | -7.53 | -8.58 | -8.51 | 69.5% | 75.8% | 0.52 | 0.53 | **0.62** | **0.61** | 0.73 | 0.73 |

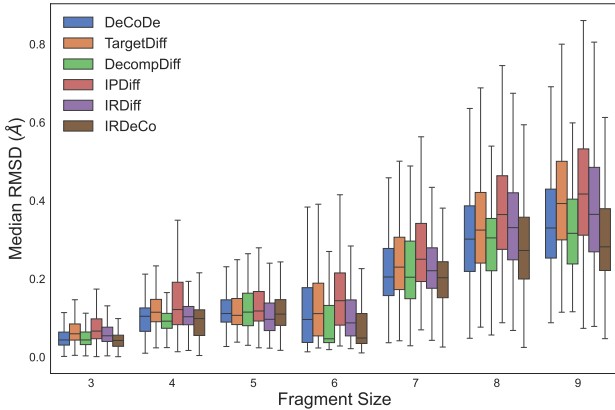

*Figure 5.* Median RMSD for rigid fragment before and after the force-field optimization.

geometries. Table 1 further confirms that IRDeCo achieves state-of-the-art performance across almost all bond types. Notably, integrating our decoupled framework consistently improves the structural fidelity of the base models, reducing the average JSD of TargetDiff and IRDiff by 18%. The bottom row of Figure 4 visually demonstrates that the distance distributions learned by DeCoDe and IRDeCo align more closely with the ground-truth than their base counterparts. For instance, for the critical mid-range carbon-carbon bonds (1.3-1.5 Å), IRDeCo depicts the concave curve of this interval preferably, while IRDiff fails. And IRDeCo significantly reduces the JSD by **48%** compared to IRDiff (0.266 → 0.137). This demonstrates that our decoupled diffusion strategy leads to more stable and physically plausible 3D conformations.

Beyond pairwise distances, a more challenging test for a 3D generative model is its ability to produce internally consistent rigid substructures such as aromatic rings, which should be planar. We measure this by performing a Merck Molecu-

lar Force Field (MMFF) (Halgren, 1996) optimization on the generated molecules and calculating the coordinate RMSD for predefined rigid fragments before and after optimization. A lower value indicates the initial generation was already close to the local energy minimum, i.e., more geometrically stable. As shown in Figure 5, models enhanced with De-CoDe show smaller median RMSD values, especially for larger rigid fragments (size ≥ 7). This demonstrates that our method not only gets inter-atomic distances right on average but also coordinates multiple atoms coherently to form stable, low-strain 3D shapes, a key requirement for generating plausible bioactive conformers.

| Methods | Vina Score (↓) | | Vina Min (↓) | | Vina Dock (↓) | | High Affinity (↑) | |
| --- | --- | --- | --- | --- | --- | --- | --- | --- |
| | Avg. | Med. | Avg. | Med. | Avg. | Med. | Avg. | Med. |
| IRDiff | -6.03 | -6.89 | -7.27 | -7.37 | -8.42 | -8.42 | 67.4% | 72.7% |
| IRDeCo | -6.69 | -7.08 | -7.54 | -7.53 | -8.58 | -8.51 | 69.9% | 75.8% |
| IRDeCo(DPO) | **-7.20** | -7.46 | **-8.14** | -8.10 | -8.88 | **-8.82** | **75.3%** | **85.8%** |
| AliDiff | -7.07 | **-7.95** | -8.09 | **-8.17** | **-8.90** | -8.81 | 73.4% | 81.4% |

*Table 3.* IRDeCo with soft-DPO fine-tuning. (↑) / (↓) denotes a larger / smaller number is better. Top 2 results are highlighted with **bold text** and underlined text, respectively.

**Target Binding Affinity.** Table 2 shows that DeCoDe-based models achieve competitive binding affinity scores compared to strong baselines. However, the improvement over the base models (TargetDiff, IRDiff) is not as pronounced as in structural metrics. This suggests that enhanced structural realism does not automatically translate to proportionally higher binding affinity, a phenomenon also noted in prior work (Gu et al., 2024). It is important to note that leading baselines like AliDiff and IPDiff explicitly incorporate affinity-oriented enhancements (e.g., DPO fine-tuning, affinity-aware pretraining), whereas DeCoDe is a *fundamental redesign of the diffusion process* without such specific optimizations. Nevertheless, superior structural

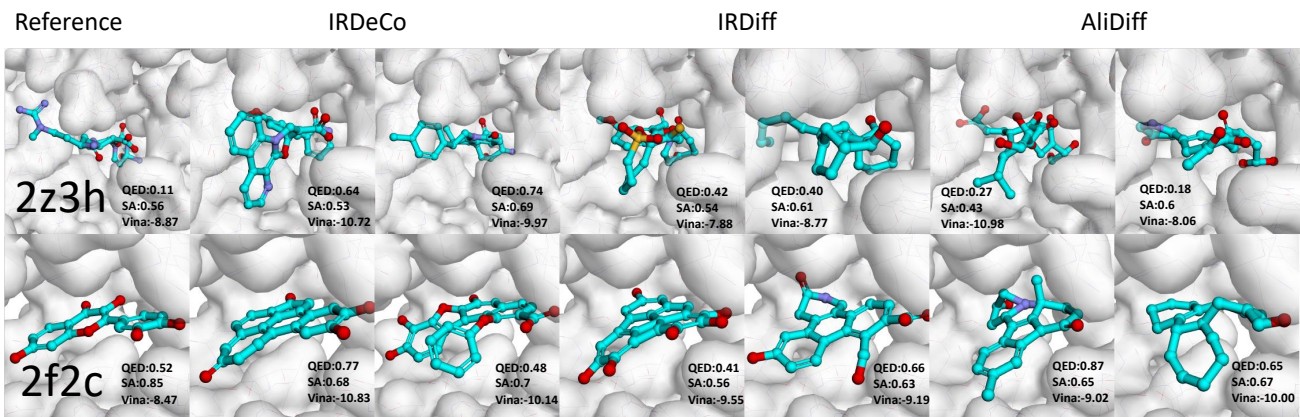

*Figure 6.* Visualizations of reference molecules and generated ligands for protein pockets (2z3h, 2f2c) from IRDeCo, IRDiff and AliDiff.

quality provides a better foundation for subsequent affinity optimization. Indeed, as shown in Table 3, when we apply the same soft-DPO fine-tuning used in AliDiff to IRDeCo, the resulting model (IRDeCo-DPO) surpasses AliDiff on most affinity metrics, demonstrating the potential synergy between our method and affinity-focused techniques.

**Molecule Properties.** DeCoDes maintain drug-likeness (QED) comparable to baselines while achieving the best synthetic accessibility (SA) score. The improved SA may stem from the more stable and realistic structures generated by our decoupled approach, which are less likely to contain strained or synthetically challenging geometries. While QED and SA are often used as coarse filters in early-stage discovery, their favorable values here indicate that DeCoDe generates practical and synthesizable candidates.

Figure 6 shows some examples of generated ligand molecules and their properties. The molecules generated by our IRDeCo have valid structures and reasonable binding poses to the target, which are supposed to be promising candidate ligands.

### 5.3. Ablation Study

**Effect of DeCoDe.** We ablate the core components of DeCoDe starting from TargetDiff (Table 4). Merely decoupling the diffusion processes of the binding position and the molecular structure without matched parameterization decoupling (*TargetDe*), degrades performance, due to the mismatch between the diffusion process and the model architecture. Introducing parameterization decoupling (*TargetDeDe*) recovers and improves performance, validating the necessity of aligned diffusion and model design. Finally, incorporating our time-dependent noise scheduling yields the full *DeCoDe* model, achieving the best overall results. This stepwise improvement confirms the effectiveness of each proposed component. The result show as Table 4:

| Methods | Vina Score (↓) | | Vina Min (↓) | | QED (↑) | | JSD (↓) | |
|---|---|---|---|---|---|---|---|---|
| | Avg. | Med. | Avg. | Med. | Avg. | Med. | C-C. | All. |
| TargetDiff | -5.47 | -6.30 | -6.64 | **-6.83** | **0.48** | **0.48** | 0.225 | 0.089 |
| TargetDe | -5.42 | -5.97 | -6.32 | -6.32 | 0.45 | 0.45 | 0.292 | 0.121 |
| TargetDeDe | -5.89 | -6.25 | -6.69 | -6.80 | 0.47 | 0.46 | 0.214 | 0.076 |
| DeCoDe | **-6.19** | **-6.45** | **-6.81** | -6.78 | **0.48** | **0.48** | **0.164** | **0.060** |

*Table 4.* Impact of Diffusion Decoupling. (↑) / (↓) denotes a larger / smaller number is better. The best results are highlighted with **bold text**.

**Impact of Different Noise Scheduling Strategy.** Our time-dependent scheduling prioritizes perturbing the global position later, guiding the model to first coarsely locate the ligand. We test three variants (Table 5): 1) *Synchronous* (same schedule for the binding position and the molecular structure), 2) *DeCoDe* (our schedule), and 3) *Inverse-DeCoDe* (swapped schedules, destroying the binding position faster than the molecular structure). *DeCoDe* performs best overall. The *Inverse* strategy performs the worst, particularly on binding affinity metrics. This can be explained: by destroying the global position signal early, the model must commit to a detailed molecular structure without knowing its final location, making it difficult for the structure to later adapt to the shape and chemistry of the protein pocket. This validates our design intuition: coarse positioning should precede structural refinement.

| Methods | Vina Score (↓) | | Vina Min (↓) | | QED (↑) | | JSD (↓) | |
|---|---|---|---|---|---|---|---|---|
| | Avg. | Med. | Avg. | Med. | Avg. | Med. | C-C. | All. |
| Synchronous | -5.92 | -6.21 | -6.72 | -6.65 | 0.48 | 0.48 | 0.176 | 0.065 |
| Invers-DeCoDe | -5.51 | -5.87 | -6.03 | -6.35 | 0.48 | 0.47 | 0.187 | 0.068 |
| DeCoDe | **-6.19** | **-6.45** | **-6.81** | **-6.78** | **0.48** | **0.48** | **0.164** | **0.060** |

*Table 5.* Influence of different Noise Scheduling. (↑) / (↓) denotes a larger / smaller number is better. The best results are highlighted with **bold text**.

For an intuitive comparison, we illustrate the generative process of DeCoDe and TargetDiff in Figure 7. It can be observed: initially, atoms are distributed chaotically near

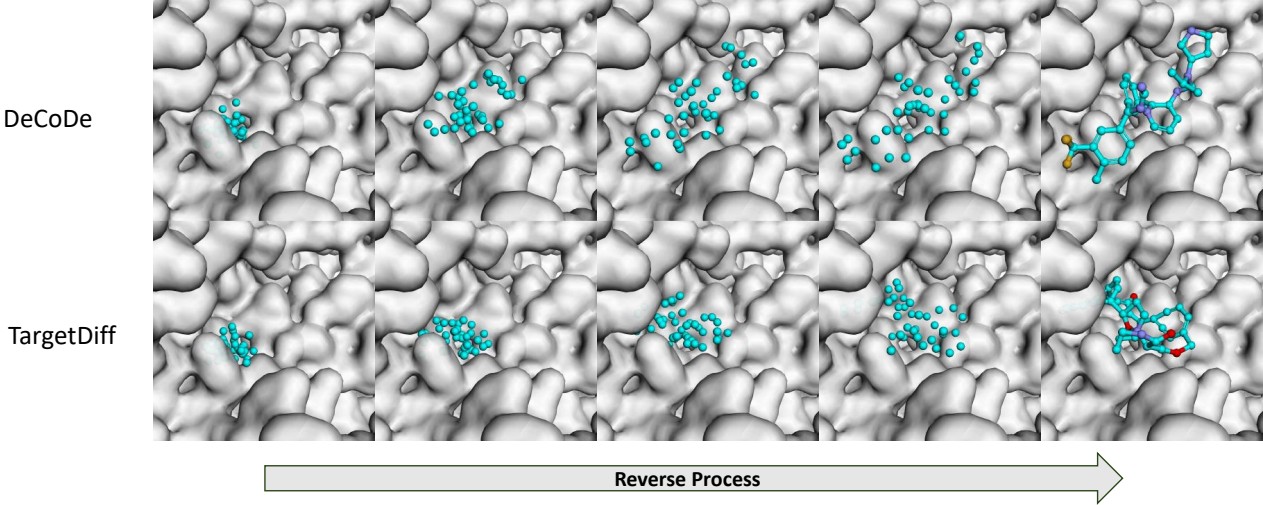

*Figure 7.* Visualizations of generation processes from DeCoDe and TargetDiff, atomic types are omitted until generation is complete.

the pocket's center of mass, with some clashes against the protein surface. As the reverse diffusion proceeds, DeCoDe first shifts atoms toward plausible binding positions and then arranges them into a well-adapted ligand molecule that fills the pocket. In contrast, TargetDiff addresses the two sub-problems simultaneously, which leads to a sub-optimal molecular geometry that does not fully occupy the pocket.

## 6. Conclusion

In this work, we propose DeCoDe, a novel diffusion framework that explicitly decouples the diffusion of global binding position and intrinsic molecular conformation for Structure-Based Drug Design. We further design a adapted time-dependent noise scheduling strategy that orchestrates an asynchronous denoising process, prioritizing position coarse-localization followed by conformational refinement, which is both efficient and intuitive. Extensive experiments on the CrossDocked2020 benchmark demonstrate the effectiveness of our method. By integrating with DeCoDe, TargetDiff and IRDiff improve their structural fidelity of the generated ligands by an average of 18%, achieving the state-of-the-art molecular structure generating, while maintaining comparable binding affinity and overall molecular properties compared to state-of-the-art baselines.

## Acknowledgements

This work was supported in part by the Advanced Materials-National Science and Technology Major Project under Grant 2024ZD0608200, in part by the National Natural Science Foundation of China under Grant 82474394, in part by Sichuan Science and Technology Program under Grant 2026NSFSC1462, and in part by the MOE (Ministry of Education in China) Liberal arts and Social Sciences Foundation under Grant 24XJCZH004.

## Impact Statement

This paper presents work whose goal is to advance the field of Machine Learning. There are many potential societal consequences of our work, none which we feel must be specifically highlighted here.

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

# A. Proof of Invariant Likelihood

As discussed in Guan et al. (2023a), an invariant initial density composed with an equivariant transition function yields an invariant distribution. Concretely, the model must satisfy:

$$p(\mathbf{X}_T, \mathbf{X}_P) = p(T_g(\mathbf{X}_T, \mathbf{X}_P))$$
$$p(\mathbf{X}_{t-1}|\mathbf{X}_t, \mathbf{X}_P) = p(T_g(\mathbf{X}_{t-1})|T_g(\mathbf{X}_t, \mathbf{X}_P)),$$

where $T_g$ denotes an SE(3)-transformation, explicitly written as $T_g(\mathbf{X}) = \boldsymbol{R}\mathbf{X} + \boldsymbol{b}$, with rotation matrix $\boldsymbol{R} \in \mathbb{R}^{3\times3}$ and translation vector $\boldsymbol{b} \in \mathbb{R}^3$.

**Invariant initial density.**  Assuming the diffusion step $T$ is sufficiently large, and following the common practice of centering the protein pocket at the origin to obtain $\hat{\mathbf{X}}_P$, the distribution of the structure component $\mathbf{X}_T^s$ given $\hat{\mathbf{X}}_P$ is a degenerate Gaussian ( CoM is strictly 0) with variance $(1 - \frac{1}{N_M})\boldsymbol{I}$, while the binding position component $\mathbf{x}_T^c$ given $\hat{\mathbf{X}}_P$ follows $\mathcal{N}(\mathbf{0}, \frac{1}{N_M}\boldsymbol{I})$. Since $\mathbf{X}_T = \mathbf{X}_T^s + \mathbf{x}_T^c$, the additivity of Gaussian distributions implies that $q(\mathbf{X}_T|\hat{\mathbf{X}}_P)$ is a standard isotropic Gaussian $\mathcal{N}(\mathbf{0}, \boldsymbol{I})$. In the generative process, we sample $\mathbf{X}_T$ from $p(\mathbf{X}_T, \hat{\mathbf{X}}_P)$, which we also set as a standard Gaussian. The standard Gaussian is isotropic and invariant under any SE(3)-transformation. Hence, $p(\mathbf{X}_T, \hat{\mathbf{X}}_P)$ is an invariant initial density.

**Equivariant transition function.**  Recall from Equation (9) in the main text that we update atom embeddings $\mathbf{H}$ and coordinates $\mathbf{X}$ alternately as:

$$\mathbf{h}_i^{l+1} = \mathbf{h}_i^l + \sum_{j\in\mathcal{N}_i} f_h^l\left(d_{ij}^l, \mathbf{h}_i^l, \mathbf{h}_j^l, \mathbf{e}_{ij}; \theta_h\right)$$

$$\Delta\mathbf{x}_i = \sum_{j\in\mathcal{N}_i} \left(\mathbf{x}_i^l - \mathbf{x}_j^l\right) \cdot f_x^l\left(d_{ij}^l, \mathbf{h}_i^{l+1}, \mathbf{h}_j^{l+1}, \mathbf{e}_{ij}; \theta_x\right)$$

$$\overline{\Delta\mathbf{x}}_{\text{protein}} = \frac{1}{N_P}\sum_{k\in\mathcal{P}} \Delta\mathbf{x}_k$$

$$\mathbf{x}_i^{l+1} = \begin{cases} \mathbf{x}_i^l + \Delta\mathbf{x}_i, & \text{for} \quad i \in \mathcal{M} \\ \mathbf{x}_i^l + \overline{\Delta\mathbf{x}}_{\text{protein}}, & \text{for} \quad i \in \mathcal{P} \end{cases}$$

After the last layer, we shift the entire complex so that the protein's center of mass returns to the origin:

$$\Delta\mathbf{x}_{\text{protein}} = \frac{1}{N_P}\sum_{k\in\mathcal{P}} \mathbf{x}_k^L$$

$$\hat{\mathbf{X}}_0 = \mathbf{X}^L - \Delta\mathbf{x}_{\text{protein}}$$

$$\hat{\mathbf{x}}_0^c = \mathbf{x}^c - \Delta\mathbf{x}_{\text{protein}}$$

$$\hat{\mathbf{X}}_0^s = \hat{\mathbf{X}}_0 - \hat{\mathbf{x}}_0^c.$$

This procedure is equivalent to keeping the protein static and updating ligand coordinates at each layer $l$ as:

$$\mathbf{x}_i^{l+1} = \begin{cases} \mathbf{x}_i^l + \Delta\mathbf{x}_i - \overline{\Delta\mathbf{x}}_{\text{protein}}, & \text{for} \quad i \in \mathcal{M} \\ \mathbf{x}_i^l, & \text{for} \quad i \in \mathcal{P} \end{cases} \tag{13}$$

Because the inter-atomic distances $d_{ij}$, edge features $e_{ij}$, and hidden features $\mathbf{h}$ are invariant under SE(3) transformations, the updates $\Delta\mathbf{x}_i$ and $\overline{\Delta\mathbf{x}}$ are thus equivariant. Consequently, the coordinate update defined in Equation (13) is SE(3)-equivariant:

$$\phi_\theta(T(\mathbf{X}^l)) = T(\mathbf{x}_i^l) + T(\Delta\mathbf{x}_i) - T(\overline{\Delta\mathbf{x}}_{\text{protein}})$$
$$= \boldsymbol{R}\mathbf{x}_i^l + \boldsymbol{b} + \boldsymbol{R}\Delta\mathbf{x}_i - \boldsymbol{R}\overline{\Delta\mathbf{x}}_{\text{protein}}$$
$$= \boldsymbol{R}\left(\mathbf{x}_i^l + \Delta\mathbf{x}_i - \overline{\Delta\mathbf{x}}_{\text{protein}}\right) + \boldsymbol{b}$$
$$= \boldsymbol{R}\mathbf{x}_i^{l+1} + \boldsymbol{b}$$
$$= T(\phi_\theta(\mathbf{X}^l)).$$

Stacking $L$ such equivariant layers yields a network whose output $\hat{\mathbf{X}}_0$ is SE(3)-equivariant w.r.t the input $\mathbf{X}_t$. The structure component $\hat{\mathbf{X}}_0^S$ is correspondingly equivariant w.r.t $\mathbf{X}_t^S$ and the binding position component $\hat{\mathbf{x}}_0^c$ is equivariant w.r.t $\mathbf{x}_t^c$. In the decoupled generative process, the mean of the reverse transition distribution $p(\hat{\mathbf{X}}_{t-1}|\mathbf{X}_t, \hat{\mathbf{X}}_P)$ is obtained as:

$$\hat{\mathbf{X}}_{t-1} = \hat{\mathbf{X}}_{t-1}^s + \hat{\mathbf{x}}_{t-1}^c$$
$$= \frac{\sqrt{\bar{\alpha}_{t-1}^s}\beta_t^s}{1-\bar{\alpha}_t^s}\hat{\mathbf{X}}_0^s + \frac{\sqrt{\bar{\alpha}_t^s}(1-\bar{\alpha}_{t-1}^s)}{1-\bar{\alpha}_t^s}\mathbf{X}_t^s$$
$$+ \frac{\sqrt{\bar{\alpha}_{t-1}^c}\beta_t^c}{1-\bar{\alpha}_t^c}\hat{\mathbf{x}}_0^c + \frac{\sqrt{\bar{\alpha}_t^c}(1-\bar{\alpha}_{t-1}^c)}{1-\bar{\alpha}_t^c}\mathbf{x}_t^c,$$

where broadcasting is omitted for brevity. The structure component $\mathbf{X}^s$ is translation-invariant by construction (it lies in the zero-CoM subspace). Moreover, with the protein's CoM fixed at the origin, the binding position component $\mathbf{x}^c$ is also translation-invariant. Thus, we need only consider a pure rotation $\mathbf{R}$. Substituting $\mathbf{R}\mathbf{X}_t$ into the above expression gives:

$$\tilde{\boldsymbol{\mu}}_\theta(\mathbf{R}(\mathbf{X}_t), t) = \frac{\sqrt{\bar{\alpha}_{t-1}^s}\beta_t^s}{1-\bar{\alpha}_t^s}\mathbf{R}(\hat{\mathbf{X}}_0^s) + \frac{\sqrt{\bar{\alpha}_t^s}(1-\bar{\alpha}_{t-1}^s)}{1-\bar{\alpha}_t^s}\mathbf{R}(\mathbf{X}_t^s)$$
$$+ \frac{\sqrt{\bar{\alpha}_{t-1}^c}\beta_t^c}{1-\bar{\alpha}_t^c}\mathbf{R}(\hat{\mathbf{x}}_0^c) + \frac{\sqrt{\bar{\alpha}_t^c}(1-\bar{\alpha}_{t-1}^c)}{1-\bar{\alpha}_t^c}\mathbf{R}(\mathbf{x}_t^c)$$
$$= \mathbf{R}(\tilde{\boldsymbol{\mu}}_\theta(\mathbf{X}_t^s, t)) + \mathbf{R}(\tilde{\boldsymbol{\mu}}_\theta(\mathbf{x}_t^c, t))$$
$$= \mathbf{R}(\tilde{\boldsymbol{\mu}}_\theta(\mathbf{X}_t, t)).$$

Hence, the reverse transition distribution $p(\mathbf{X}_{t-1}|\mathbf{X}_t, \hat{\mathbf{X}}_P)$ is equivariant.

**Proof of invariant likelihood.** Combining the invariant initial density and the equivariant transition function, the model's likelihood satisfies:

$$p_\theta(T_g(\mathbf{X}_0, \hat{\mathbf{X}}_P)) = \int p(T_g(\mathbf{X}_T, \hat{\mathbf{X}}_P)) \sum_{t=1}^T p_\theta(T_g(\mathbf{X}_{t-1})|T_g(\mathbf{X}_t, \hat{\mathbf{X}}_P)))$$
$$= \int p(\mathbf{X}_T, \hat{\mathbf{X}}_P) \sum_{t=1}^T p_\theta(T_g(\mathbf{X}_{t-1})|T_g(\mathbf{X}_t, \hat{\mathbf{X}}_P))) \qquad \leftarrow \text{Invariant Prior}$$
$$= \int p(\mathbf{X}_T, \hat{\mathbf{X}}_P) \sum_{t=1}^T p_\theta(\mathbf{X}_{t-1}|\mathbf{X}_t, \hat{\mathbf{X}}_P) \qquad \leftarrow \text{Equivariant Transition}$$
$$= p_\theta(\mathbf{X}_0, \hat{\mathbf{X}}_P)$$

Therefore, the likelihood $p_\theta(\mathbf{X}_0|\hat{\mathbf{X}}_P)$ is invariant under any SE(3)-transformation of the input complex, as required.

# B. Derivation of Loss Weighting Coefficient

This appendix provides the derivation for the time-dependent weighting coefficient $\gamma_t = \frac{\bar{\alpha}_t}{1-\bar{\alpha}_t}$ used in the training objectives Equation (11). The result establishes the equivalence between the clean-data prediction loss and a reweighted noise-prediction loss, motivating our design choice.

Recall the forward diffusion process for a generic coordinate variable $\mathbf{X}$ (which applies to both $\mathbf{X}^s$ and $\mathbf{x}^c$ in our decoupled framework):

$$\mathbf{X}_t = \sqrt{\bar{\alpha}_t}\mathbf{X}_0 + \sqrt{1-\bar{\alpha}_t}\epsilon_t, \qquad \epsilon \sim \mathcal{N}(\mathbf{0}, \mathbf{I}) \tag{14}$$

where $\bar{\alpha}_t = \prod_{s=1}^t (1-\beta_s)$ and $\{\beta\}$ defines the noise schedule. Let the model's prediction of the clean data be denoted by $\hat{\mathbf{X}}_0 = \phi_\theta(\mathbf{X}_t, t)$, From the forward process, the corresponding prediction of the noise $\epsilon$ is obtained by rearranging the above relation:

$$\hat{\epsilon} = \frac{\mathbf{X}_t - \sqrt{\bar{\alpha}_t}\hat{\mathbf{X}}_0}{\sqrt{1-\bar{\alpha}_t}} \tag{15}$$

The loss adopted by (Ho et al., 2020) are:

$$\mathbb{E}_{\mathbf{X}_0,\epsilon,t}[\|\epsilon - \hat{\epsilon}\|^2] = \mathbb{E}_{\mathbf{X}_0,\epsilon,t}[\frac{\bar{\alpha}_t}{1 - \bar{\alpha}_t}\left\|\mathbf{X}_0 - \hat{\mathbf{X}}_0\right\|^2] \tag{16}$$

Thus, minimizing the mean squared error in predicting the clean data $\mathbf{X}_0$ is equivalent to minimizing a time-weighted version of the noise-prediction loss, with the weighting coefficient:

$$\gamma_t = \frac{\bar{\alpha}_t}{1 - \bar{\alpha}_t}$$

## C. Overall Training and Sampling Procedures

We summarize the overall training and sampling procedures of DeCoDe as Algorithms 1 and 2, respectively.

---

**Algorithm 1** Training Procedure of DeCoDe

---

**Input:** Protein-ligand binding dataset $\{\mathcal{P}, \mathcal{M}\}_{i=1}^N$, learnable diffusion denoising model $\phi_\theta$
1: **while** $\phi_\theta$ not converge **do**
2:   Sample diffusion time $t \in \mathcal{U}(0, \ldots, T)$
3:   Move the complex to make CoM of protein atoms zero
4:   Decompose ligand molecule $\mathbf{X}$ into $\mathbf{X}^s, \mathbf{x}^c$, and Gaussian noise $\epsilon$ into $\epsilon^s, \epsilon^c$, where $\epsilon \in \mathcal{N}(\mathbf{0}, \boldsymbol{I})$     Equation (5)
5:   Perturb $\mathbf{X}_0^s$ to obtain $\mathbf{X}_t^s$: $\mathbf{X}_t^s = \sqrt{\bar{\alpha}_t^s}\mathbf{X}_0^s + (1 - \bar{\alpha}_t^s)\epsilon^s$,
6:   Perturb $\mathbf{x}_0^c$ to obtain $\mathbf{x}_t^c$: $\mathbf{x}_t^c = \sqrt{\bar{\alpha}_t^c}\mathbf{x}_0^c + (1 - \bar{\alpha}_t^c)\epsilon^c$,
7:   Perturb $\mathbf{V}_0$ to obtain $\mathbf{V}_t$:
     $\log \boldsymbol{c} = \log(\bar{\alpha}_t \mathbf{V}_0 + (1 - \bar{\alpha}_t)/K)$
     $\mathbf{V}_t = \texttt{one\_hot}(\arg\max_i[g_i + \log c_i])$, where $g \sim \text{Gumbel}(0, 1)$
8:   Reconstruct full coordinates of ligand atoms $\mathbf{X}_t$: $\mathbf{X}_t = \mathbf{X}_t^s + \mathbf{1}(\mathbf{x}_t^c)^\top$
9:   Computing inter-atomic interactions     Equation (9)
10:   Move the complex so that protein's CoM at the origin and decompose $\hat{\mathbf{X}}_0$ into $(\hat{\mathbf{X}}_0^s, \hat{\mathbf{x}}_0^c)$     Equation (10)
11:   Predict $\hat{\mathbf{V}}_0 = SoftMax(MLP(\mathbf{H}^L))$
12:   Compute loss $L$ with $(\hat{\mathbf{X}}_0^s, \hat{\mathbf{x}}_0^c, \hat{\mathbf{V}}_0)$ and $(\mathbf{X}_0^s, \mathbf{x}_0^c, \mathbf{V}_0)$     Equations (11) and (12)
13:   Update $\theta$ by minimizing $L$
14: **end while**

---

**Algorithm 2** Sampling Procedure of DeCoDe

---

**Input:** The protein binding site $\mathcal{P}$, the learned denoising model $\phi_\theta$.
**Output:** Generated ligand molecule $\mathcal{M}$ that binds to the protein pocket.
1: Sample the number of atoms in $\mathcal{M}$ based on a prior distribution conditioned on the pocket size
2: Move CoM of protein atoms to zero
3: Sample initial molecular atom coordinates $\mathbf{X}_T$ and atom types $\mathbf{V}_T$:
   $\mathbf{X}_T \in \mathcal{N}(0, \boldsymbol{I})$
   $\mathbf{V}_T = \texttt{one\_hot}(\arg\max_i g_i)$, where $g \sim \text{Gumbel}(0, 1)$
4: Decompose ligand molecule $\mathbf{X}_T$ into $\mathbf{X}_T^s, \mathbf{x}_T^c$     Equation (5)
5: **for** $t$ in $T, T-1, \ldots, 1$ **do**
6:   Predict $[\hat{\mathbf{X}}_0^s, \hat{\mathbf{x}}_0^c, \hat{\mathbf{V}}_0]$ from $[\mathbf{X}_t^s, \mathbf{x}_t^c, \mathbf{V}_t]$ with $\phi_\theta$:
7:   Sample $\mathbf{X}_{t-1}^s$ from the posterior $p_\theta(\mathbf{X}_{t-1}^s|\mathbf{X}_t^s, \hat{\mathbf{X}}_0^s)$     Equation (7)
8:   Sample $\mathbf{x}_{t-1}^c$ from the posterior $p_\theta(\mathbf{x}_{t-1}^c|\mathbf{x}_t^c, \hat{\mathbf{x}}_0^c)$     Equation (7)
9:   Sample $\mathbf{V}_{t-1}$ from the posterior $p_\theta(\mathbf{V}_{t-1}|\mathbf{V}_t, \hat{\mathbf{V}}_0)$     Equation (7)
10: **end for**
11: Reconstruct full coordinates of ligand atoms $\mathbf{X}_0$: $\mathbf{X}_0 = \mathbf{X}_0^s + \mathbf{1}(\mathbf{x}_0^c)^\top$
12: Return $\mathcal{M}$: $[\mathbf{X}_0, \mathbf{V}_0]$

---

## D. Implementation of Noise Schedule

Peng et al. (2023) introduce an adaptive noise schedule based on the sigmoid function, referred to as the "advance schedule". It is parameterized by three hyper-parameters $s_1, s_T$ and $w$ which allow the schedule curve to be flexibly adjusted. The schedule is defined as:

$$
\begin{aligned}
s &= (s_T - s_1)/(\text{sigmoid}(-w) - \text{sigmoid}(w)) \\
b &= 0.5 \times (s_1 + s_T - s) \\
\bar{\alpha}_t &= s \times \text{sigmoid}(-w(2t/T - 1)) + b.
\end{aligned}
\tag{17}
$$

Here $s_1$ and $s_T$ set the endpoints $\bar{\alpha}_1$ and $\bar{\alpha}_T$, thereby controlling the overall range of noise added during the diffusion process. The parameter $w$ governs the slope of the schedule in the mid-range steps, determining how rapidly $\bar{\alpha}_t$ changes at the beginning and end of the process.

In our implementation, we choose the parameters $s_1 = 0.9999, s_T = 0.0001, w = 4$ for the molecular structure diffusion. For the binding position diffusion, we employ a modified version that uses only the first half of the schedule, which we refer to as the "**half-advance schedule**":

$$
\begin{aligned}
s &= (s_T - s_1)/(\text{sigmoid}(0) - \text{sigmoid}(w)) \\
b &= s_T - 0.5 \times s \\
\bar{\alpha}_t &= s \times \text{sigmoid}(-w(t/T - 1)) + b.
\end{aligned}
\tag{18}
$$

We set $s_1 = 0.9999, s_T = 0.0001$ and $w = 4$.

## E. Experiment Details

### E.1. Training Details

We employ the Adam optimizer (Kingma & Ba, 2014) with a learning rate of $1e-3$, $betas = (0.95, 0.999)$, a batch size of 4, and a gradient norm clipped at 8. The learning rate is decayed exponentially with a factor of 0.6, subject to a minimum of $1e-6$, and is reduced when the validation loss shows no improvement for 10 consecutive evaluations. Following the practice in their original papers, we balance the atom type loss and atom position loss by applying a scaling factor $\lambda = 100$ to the atom type loss. Training is performed on a single NVIDIA RTX 4090 48GB GPU, and the model typically converges within 36 hours and 600k steps.

### E.2. Additional Results

To further validate the effectiveness of our method, we report additional structural metrics. Specifically, we compute the average Jensen–Shannon divergence (JSD) between the bond-angle and torsion-angle distributions of the generated molecules and those of the reference; We also report the PoseBusters (Buttenschoen et al., 2024) passing rate (PB-Valid), which aggregates 17 physical-chemical and steric tests and provides a more comprehensive assessment of structural plausibility. The result is as follows.

*Table 6.* Average Jensen-Shannon divergence of bond (torsion)angle distributions, and PB-Valid. (↑) / (↓) denotes a larger / smaller number is better. DeCoDe is based on TargetDiff, and IRDeCo(De) corresponds to IRDiff.

| Metrics | TargetDiff | DeCoDe | IRDiff | IRDeCo | DecompDiff | IPDiff |
|---|---|---|---|---|---|---|
| PB-Valid (↑) | 50.5% | 64.6% | 41.4% | 73.2% | 68.8% | 38.1% |
| Bond Angle (↓) | 0.44 | 0.41 | 0.46 | 0.38 | 0.41 | 0.48 |
| Torsion Angle (↓) | 0.41 | 0.39 | 0.44 | 0.36 | 0.37 | 0.44 |

