# OpenReview forum: "DeCoDe: Decoupling Binding Position and Molecular Conformation in 3D Ligand Diffusion for Structure-Based Drug Design"
_ICML.cc/2026/Conference — ICML 2026 spotlight_

### Official Review · Reviewer_Q4Xo · 2026-03-09

**Soundness:** 3
**Presentation:** 3
**Significance:** 3
**Originality:** 3
**Overall Recommendation:** 4
**Confidence:** 3

**Summary:**

This paper proposes the DecoDe framework, which explicitly decouples ligand coordinates into global binding position and internal molecular conformation, and designs independent diffusion processes for each component. By introducing a time-dependent asynchronous noise scheduling strategy, the proposed method guides the reverse generation process to first perform coarse ligand placement within the protein pocket, followed by progressive refinement of the molecular structure, thereby improving generation efficiency and stability. Experimental results demonstrate that integrating DecoDe into TargetDiff and IRDiff maintains competitive performance with existing methods in terms of structural fidelity and binding affinity.

**Compliance With Llm Reviewing Policy:**

Affirmed.

**Final Justification:**

The rebuttal has addressed my main concerns. I maintain the original positive score.

**Key Questions For Authors:**

1. If the pocket position is not marked and the whole protein is used as input instead, can this method be extended to simultaneously handle both binding site prediction and ligand structure generation?

2. Could you please add visualizations of the generation process to demonstrate the actual generation of some molecules, thereby highlighting the characteristics of the proposed noise scheduling?

**Limitations:**

No. Authors are suggested to discuss their limitations, such as the diversity of designed ligands can be improved.

**Strengths And Weaknesses:**

Strengths
- The method effectively decouples ligand coordinates into the global binding position via parameterization decoupling, and the proposed time-dependent noise scheduling presents a sound and novel design. The corresponding experimental results are well-supported.
- The writing of the manuscript is clear and coherent.

Weaknesses

1. Although the paper claims efficient generation, it lacks systematic computational comparisons, including training time, sampling time/steps, convergence speed, and computational overhead. This makes it difficult to fully validate the claimed efficiency advantage.
2. Regarding reproducibility, the manuscript should supplement detailed experimental settings, such as hyperparameter settings, training cost of all experiments, and introduction of data and algorithm pseudocode of the DPO experiment. Publicly releasing the source code would also be strongly recommended.

---

> ### Author Rebuttal · Authors · 2026-03-31
>
> We sincerely thank the reviewer for the positive recommendation and the valuable, constructive feedback. Your suggestions are excellent and will undoubtedly strengthen our paper. Our point-by-point responses and planned revisions are detailed below.
>
> ---
>
> **W1. On the lack of systematic computational comparisons.**
>
> Thank you for this important point. A comprehensive evaluation should indeed cover training, inference, and convergence. We will provide a full comparison in the revised manuscript:
>
> - Inference Time/Steps:​ As noted in our response to **Reviewer Dcjo's Q3**, DeCoDe's framework does not modify the backbone sampler; therefore, its per-molecule inference time and sampling steps are virtually identical to the base model.
>
> - Training Cost & Convergence:​ The decoupling operation adds negligible parameters. Consequently, the training memory footprint, parameter count, and required epochs are comparable to those of the backbone models. We will add the training loss curves to the Appendix.
>
> - Clarification on "Efficiency":​ We wish to clarify that the "efficiency" claimed in our work refers to the **efficiency in exploring the molecular conformational space**​ (which our experiments demonstrate), not the raw speed of training or inference. Our framework builds upon existing diffusion backbones and does not alter their fundamental training or inference efficiency. We will revise the manuscript to describe this contribution more precisely, focusing on its role in enabling effective sample-space exploration.
>
> ---
>
> **W2. On detailed experimental settings and reproducibility.**
>
> We completely agree on the importance of reproducibility. We will take the following actions:
>
> - Detailed Hyperparameters: We will expand a "Experimental Setup" section in Appendix to list all key hyperparameters.
>
> - DPO Details: The DPO fine-tuning strategy and parameters closely follow the established implementation in [1]. To maintain focus on our core contributions, we will reference [1] for methodological details and provide the specific configurations used in our experiments in the Appendix to ensure full reproducibility.
>
> - Code & Model Release: We commit to releasing the full code, model weights, and detailed documentation on GitHub upon acceptance.
>
> ---
>
> **Q1. If the pocket position is not marked and the whole protein is used as input instead, can this method be extended to simultaneously handle both binding site prediction and ligand structure generation?**
>
> This is an insightful question. The DeCoDe framework does not inherently require a pre-marked pocket as input. The model learns to infer the binding position from the protein context during training. This is enabled because the position diffusion process can be conditioned on the entire protein structure, while the independent structure diffusion remains unaffected. This decoupling is a key advantage of our framework, providing a foundation for potential extension to joint binding-site prediction and generation.
>
> ---
>
> **Q2. Could you please add visualizations of the generation process to demonstrate the actual generation of some molecules, thereby highlighting the characteristics of the proposed noise scheduling?**
>
> Thank you for this excellent suggestion. We have created visualizations to illustrate the generation process, available at https://anonymous.4open.science/r/anonym056P/visualized_generation_process.pdf.
>
> The visuals clearly demonstrate that DeCoDe first guides the ligand's point cloud to the pocket region before refining its detailed structure. In contrast, a coupled diffusion baseline (TargetDiff) attempts to generate structure too early, leading to suboptimal placement and clashes with the protein's surface. This effectively highlights the benefit of our time-dependent noise scheduling. We will include a key figure in the manuscript's Appendix to illustrate this.
>
> ---
>
> **Limitations Discussion**:
>
> We appreciate the reviewer's suggestion to discuss limitations. While **DeCoDe performs on par with baselines on the Diversity metrics**, we acknowledge that achieving high diversity in generative models remains a general challenge. The focus of our current work is on improving the realism and binding affinity of generated molecules within a decoupled generation framework. Exploring advanced sampling techniques or explicit objectives to further enhance the diversity and novelty of outputs is a valuable direction for future work, and we will include this discussion in the revised manuscript.
>
> ---
>
> We are grateful for the reviewer's time and insightful comments, which have greatly helped us improve the work. We believe the planned revisions will fully address the points raised.
>
> ---
>
> [1] Gu, S., Xu, M., Powers, A., Nie, W., Geffner, T., Kreis,K., Leskovec, J., Vahdat, A., and Ermon, S. Aligning target-aware molecule diffusion models with exact energy optimization. In The Thirty-eighth Annual Conference on Neural Information Processing Systems, 2024.

---

> > ### Author Rebuttal · Reviewer_Q4Xo · 2026-04-04
> >
> > Thank you for the response. I will maintain the positive score. I hope the authors can open-source their code soon to ensure reproducibility and make substantial contributions to the community.

---

> > > ### Author Response · Authors · 2026-04-04
> > >
> > > We sincerely thanks again for your positive recommendation and the insightful review. Wishing you a happy life.

---

### Official Review · Reviewer_Dcjo · 2026-03-10

**Soundness:** 2
**Presentation:** 3
**Significance:** 2
**Originality:** 2
**Overall Recommendation:** 4
**Confidence:** 4

**Summary:**

The authors identify a major bottleneck in existing SBDD methods that these methods generate binding position and molecular conformation simultaneously, which causes unnecessary complexity. They propose DeCoDe, a framework that explicitly decouples these two factors. Specifically, they also implement a time-dependent noise-scheduling strategy that encourages the model to first find binding region and then adjusting molecular conformation. Experiments on CrossDocked2020 show DeCoDe can boost molecular structural properties.

**Compliance With Llm Reviewing Policy:**

Affirmed.

**Final Justification:**

My concerns have been addressed. The authors added more experiments, and the results on PoseBusters are promising. They should add this to their final version, which can strengthen their claim on designing better binding positions and conformations.

**Key Questions For Authors:**

1. Can you add extra validation using PoseBusters? This would greatly enhance this work.
2. Can you select molecules with identical size (e.g., number of atoms) to compare?
3. Can you provide the time consumption of DeCoDe, especially compared to TargetDiff and IRDiff?

**Limitations:**

Yes.

**Strengths And Weaknesses:**

***Strengths***
1. The design of DeCoDe fits the claim for the necessity of decoupling binding position and molecular conformation. The design of an asynchronous time schedule is also interesting.
2. DeCoDe is a transplantable design, validated on two backbones and show universial improvements in terms of JSD. MMFF RMSD also confirms the effectiveness.
3. Experiments are strictly done with multiple metrics, covering a wide range of evalution perspectives.

***Weaknesses***
1. The core insight of decoupling binding position and molecular conformation is implicitly proposed in existing works. Although their motivation may not be identical, the implementation for decomposing molecules by the center of mass is not new.
2. Following 1, the binding box is manually set to a small region in most cases, and the conformation can be adjusted by other algorithms (e.g., Vina Dock). Does this make the core idea somehow redundant?
3. Some important baselines are missing: VoxBind[1], MolCRAFT[2]. Moreover, AliDiff is not compared in Table 1.
4. Evaluation of molecular structures can be richer. The major highlight of this work is generating molecules with better conformation and binding sites; extra validation using tools like PoseBusters[3] can greatly support this finding.
5. The selected case may not be fair. In Figure 6, the selected molecules for IRDeCo seem larger than those for IRDiff and AliDiff. This size mismatch will result in better Vina scores and worse SA scores.

[1] Pinheiro P O, Jamasb A, Mahmood O, et al. Structure-based drug design by denoising voxel grids[J]. arXiv preprint arXiv:2405.03961, 2024.

[2] Qu Y, Qiu K, Song Y, et al. MolCRAFT: structure-based drug design in continuous parameter space[J]. arXiv preprint arXiv:2404.12141, 2024.

[3] Buttenschoen M, Morris G M, Deane C M. PoseBusters: AI-based docking methods fail to generate physically valid poses or generalise to novel sequences[J]. Chemical Science, 2024, 15(9): 3130-3139.

---

> ### Author Rebuttal · Authors · 2026-03-31
>
> Thank you for your thoughtful and constructive reviews. We have carefully considered all points and provide our responses below, hoping to fully address your concerns.
>
> ---
>
> **W1. On the novelty of decoupling and centroid decomposition.**
>
> We agree that using centroid decompose is known and decoupling may be **implicit in prior work**. However, we are the **first to explicitly decouple** binding position and structure diffusion in diffusion models and propose to diffuse them asynchronously according to their distinct roles in pocket binding. This purposeful design enforces a coarse-to-fine generation order, which is our core contribution. The centroid decomposition is a methodological step to achieve diffusion decoupling rather than a claimed contribution. Moreover, our framework is end-to-end, requires no external support, and easily integrates with existing backbones (praised by **Reviewer qDVY**); these are what the prior implicit work lacks.
>
> ---
>
> **W2. On manually setting binding box.**
>
> You are concerned that our work may be made somehow redundant due to the manually-set binding box. However, we address an important limitation. Such a manual box is biased and its center does not generally coincide with a ligand's center of mass. Generating a structure directly within this box resembles a "Pocket Inpainting" task rather than the de novo design of a complete ligand molecule, which can compromise structural fidelity and even lead to incomplete structures. Our method is designed to address this limitation, and the experiment validates the rationale and effectiveness of our work.
>
> ---
>
> **W3. On baseline selection (VoxBind, MolCRAFT, AliDiff).**
>
> Thank you for pointing out these references; we have cited them in the revised manuscript. We note that our main experiments already compare against recent SOTA methods (IPDiff and AliDiff). Our work focuses on improving E(3)-equivariant point-cloud diffusion models, so our primary comparisons are within this category. VoxBind is voxel-based and MolCRAFT is a Bayesian Flow Network method; their strong performance represents a different, valuable solution for SBDD. We provide a concise comparison below (full results will be in the manuscript), confirming that our method achieves competitive or superior performance on key metrics.
>
> | Method |Bone JSD↓| Vina Score↓| Vina Dock↓|QED↑|SA↑| Div|Size|
> | ----------- | ----------- |----------- |----------- |----------- |----------- |----------- |----------- |
> | IRDeCo | **0.231** | -6.69| **-8.58**|0.52|0.62|0.73|24.5|
> | VoxBind | 0.363 | **-6.94**| -8.30|0.57|0.70|0.73|23.4|
> | MolCRAFT| 0.357 | -6.59| -7.92|0.50|0.69|0.72|22.7|
>
> - Regarding AliDiff:​ It is a DPO fine-tuning technique for affinity optimization, not a base diffusion model for structure generation. Therefore, it is omitted from Table 1 (focused on structural metrics). A full comparison in our response to **ReviewerHY4A's W1** shows our method outperforms AliDiff on structural metrics.
>
> ---
>
> **W4. On enriching molecular structure evaluation.**
>
> Excellent suggestion!  We add evaluations for bond angle distribution, torsion angle distribution, and the PoseBusters validity rate to provide a more comprehensive geometric assessment. A compact summary is below (full details in manuscript).
>
> | Metrics|TargetDiff | DeCoDe | IRDiff | IRDeCo|DecompDiff |IPDiff |
> | ----------- | ----------- |----------- |----------- |----------- |----------- |----------- |
> | PB-Valid↑|50.5%|64.6%|41.4%|73.2%|68.8%|38.1%|
> |Angel JSD↓|0.44 |0.41|0.46|0.38|0.41|0.48|
> |Torsion JSD↓ | 0.41 | 0.39|0.44|0.36|0.37|0.44|
>
> DeCoDe consistently improves the structural performance of both the TargetDiff and IRDiff backbones.
>
> ---
>
> **W5. On case fairness in Figure 6.**
>
> We apologize for the oversight in the initial version of Figure 6, which is intended for qualitative illustration only. We re-plot it using ligands of comparable size for a fair visual comparison (see https://anonymous.4open.science/r/anonym056P/figure6.pdf) and include additional examples in the Appendix. The table below confirms that DeCoDe does not generate systematically larger molecules, as the average size is similar to the backbone models and other baselines.
>
> | Metrics|TargetDiff | DeCoDe | IRDiff | IRDeCo|AliDiff |
> | ----------- | ----------- |----------- |----------- |----------- |----------- |
> | Avg Size | 24.24 | 23.15| 24.34|24.55|24.43|
>
> ---
>
> **Q1. Can you add extra validation using PoseBusters?**
>
> See W4.
>
> ---
>
> **Q2. Can you select molecules with identical size to compare?**
>
> See W5.
>
> ---
>
> **Q3. Can you provide the time consumption of DeCoDe?**
>
> The average inference time per molecule (1000 sampling steps) is as follows:
>
> | Metrics|TargetDiff | DeCoDe | IRDiff | IRDeCo|
> | ----------- | ----------- |----------- |----------- |----------- |
> | Time(s)| 13.2 | 13.3| 22.5|22.5|
>
> DeCoDe does not alter the backbone sampler; therefore, its inference time is virtually identical.
>
> ---
>
> Thanks again for the insightful feedback.

---

> > ### Author Rebuttal · Reviewer_Dcjo · 2026-04-03
> >
> > Seems fine by me, and I have raised my score. Thanks for the response.

---

> > > ### Author Response · Authors · 2026-04-03
> > >
> > > Thank you for the constructive reviews. We sincerely thank you for raising the score. Wish you a happy life.

---

### Official Review · Reviewer_qDVY · 2026-03-11

**Soundness:** 3
**Presentation:** 3
**Significance:** 3
**Originality:** 2
**Overall Recommendation:** 5
**Confidence:** 4

**Summary:**

The paper builds on prior diffusion models to advance 3D design of small molecule binders. The key modification in the proposed approach is to decouple the ligand position and its internal structure in the forward (and reverse) processes. This permits one to use different corruption schedules for position vs conformation. In particular, the proposed schedule implies that the reverse process aims to first resolve a rough binding position prior to settling on the binder specifics, which perhaps agrees better with intuition, and may simplify the overall learning problem. The decoupling can be realized through simple post modifications to existing architectures, just a few added (separated) operations for the forward and reverse processes. The authors also modify the updates so that the protein position can change and this is successively absorbed into the ligand positional update.

**Compliance With Llm Reviewing Policy:**

Affirmed.

**Final Justification:**

As already stated in the rebuttal acknowledgement, my assessment remains between weak accept (due to lack of originality) and accept (simplicity, plug and play nature of the modification). The authors' response did not resolve deeper issue about noise schedules. I understand that further modifications would potentially be required to address this.

**Key Questions For Authors:**

See above re noise schedule, dataset separation.

**Limitations:**

yes

**Strengths And Weaknesses:**

The paper is very clearly written. The proposed changes are easily adopted as they allow one to use an existing denoising architecture just with a few post-modifications. The idea that one should prioritize the corruption of the ligand internal conformation initially and then later shift towards corrupting the binding position seems natural though ideally would come with some justification other than overall performance. During generation, the model conversely needs to resolve the rough position first, then specifics.

One natural question concerns the "optimality" of the two noise schedules. How much did the authors explore different alternatives? One could (should?) instead learn a denoising model that is explicitly conditioned on the two noise scales rather than the overall synchronized time t. Currently, the denoising model only implicitly learns to understands how t relates to positional vs conformational noise levels. This does not permit one to use different schedules at inference time, nor optimize inference time schedules based on error analysis.

What is meant on line 274 that there's "no sequence overlap" to the training set? Do you mean something like "at most 30% overlap", i.e., remote homology, or what?

The results are good, comprehensive, and consistent but not substantial improvements.

Perhaps the main downside is (lack of) originality.

---

> ### Author Rebuttal · Authors · 2026-03-31
>
> We sincerely thank you for the positive recommendation and the valuable, constructive feedback. Your suggestions are excellent and will undoubtedly strengthen our paper. We address your specific points below.
>
> ---
>
> **W1. On the "optimality" of the two noise schedules.**
>
> This is an excellent and profound point. We agree that a more general framework, where the denoising network is explicitly conditioned on independent noise scales​ for position and structure, would allow for flexible adjustment or even optimization of the schedules during inference. This is a promising direction for future work.
>
> In this work, our primary goal is to propose a simple, **plug-and-play** decoupling framework without modifying the core architecture of existing diffusion models' denoising networks. Hence, we choose deterministic schedules coupled to a single timestep $t$. This design preserves simplicity and ease of use: our method can be directly applied to existing backbones without significantly altering the denoising network.
>
> We experiment with several schedule functions dependent on $t$ by adjusting the parameter $w$ of noise schedules (see comparative table below). The chosen scheme $w=4$ provided the best balance between binding affinity and structural quality.
>
> | $w$ |C-C JSD↓| Vina Score↓| Vina Dock↓|QED↑|
> | ----------- | ----------- |----------- |----------- |----------- |
> | 1 | 0.183 | -5.79|-7.68|0.47|
> | 4| **0.164**| **-6.19**|-7.79|**0.48**|
> | 7| 0.197 | -6.11| **-7.83**|0.47|
>
> We will discuss this design choice and its trade-offs more thoroughly in the revision and mention "learning conditional, adaptive schedules" as valuable future work.
>
> ---
>
> **W2. On the dataset statement (line 274).**
>
> We apologize for the imprecise phrasing. Our intended meaning is that there is no significant sequence homology​ between the proteins in the test set and those in the training set (typically using a threshold of ≤30% sequence identity), ensuring a fair evaluation of generalization. This data split is **identical to** the partitioning scheme used by **baselines**​ we compare against. We will revise the sentence to clearly state the split is based on a sequence similarity threshold to avoid any misunderstanding.
>
> ---
>
> **W3. On the perceived perhaps lack of originality.**
>
> We appreciate this perspective. While our work builds upon existing diffusion models, we believe its core contribution lies in being the first to explicitly structure the generative process by decoupling position and structure diffusion for SBDD, and to systematically design a matched time-dependent noise schedule​ for this framework. This combination is non-trivial and stems from rethinking the task's essence (the physical scale difference between binding position and molecular conformation). The experiments demonstrate that this design consistently and robustly improves different SOTA backbones, validating its value as an effective, general-purpose design. Its conceptual clarity and ease of implementation are also merits from a methodological standpoint.
>
> ---
>
> We thank you again for the thoughtful feedback, which has strengthened our paper. We hope our responses satisfactorily address your concerns. Wishing you a happy life.

---

> > ### Author Rebuttal · Reviewer_qDVY · 2026-04-03
> >
> > The authors' response does not further resolve deeper issue about noise schedules but keeps the status quo. My assessment of the paper remains between weak accept (due to lack of originality) and accept (simplicity, easily adopted by others).

---

> > > ### Author Response · Authors · 2026-04-04
> > >
> > > We sincerely thank you again for the positive recommendation and the valuable, constructive feedback. Wishing you a happy life.

---

### Official Review · Reviewer_HY4A · 2026-03-18

**Soundness:** 3
**Presentation:** 3
**Significance:** 3
**Originality:** 3
**Overall Recommendation:** 4
**Confidence:** 2

**Summary:**

The authors propose a new method for understanding ligand binding, called DeCoDe, which is implemented in two stages: Diffusion decoupling and Time-dependent diffusion scheduling. The model was tested using the CrossDocked2020 benchmarks and has outperformed existing methods in both accuracy and speed.

**Compliance With Llm Reviewing Policy:**

Affirmed.

**Key Questions For Authors:**

Overall, the paper is well-written and presented in an organized manner. However, there are a number of areas where additional clarification would be helpful (e.g., performance comparisons and method novelty). By making these suggested changes, this paper would be even stronger. To summarize, I incline towards acceptance if the above mentioned concerns are addressed.

**Limitations:**

yes

**Strengths And Weaknesses:**

Strengths
1. The proposed method offers a diffusion-based selection procedure that clearly separates the generation of binding position and molecular conformation of ligands, an important aspect of structural design.
2. The time-dependent noise-scheduling approach uses a combination of coarse-to-fine level methods for generating the ligand. The resulting ligand will have an initial approximate binding location and will then be refined according to the same steps as a human would follow in the course of drug discovery.
3. The paper is well written and organized.

Weakness
1. One of the main concerns of the proposed work is the sub-optimal performance of the proposed work compared to the different baselines. The authors should clarify this in their discussion.
2. Another concern is the limited novelty, it is not clear from the Introduction how the proposed method is different from existing methods like AliDiff or IRDiff.
3. In Figure 6, the generated ligands are not easy to visualize. The authors can refer to Figure 4 of IRDiff paper or Figure 2 of AliDiff paper.

---

> ### Author Rebuttal · Authors · 2026-03-30
>
> We appreciate the reviewer’s insightful comments and the opportunity to clarify our work. Our point-by-point responses are provided below.
>
> ---
>
> **W1. On sub-optimal performance compared to the different baselines.**
>
> Thank you for raising this point. We understand the concern regarding “sub-optimal performance” likely refers to the binding affinity metrics, where AliDiff shows an advantage. We initially addressed this briefly in the manuscript (lines 356–359) and are pleased to provide a more detailed discussion here, which will be added to the Appendix in the revised version manuscript.
>
> The performance difference stems from a fundamental divergence in objective. AliDiff is specifically designed to enhance the binding affinity of SBDD diffusion models and therefore employs Direct Preference Optimization (DPO) to explicitly optimize for that objective. In contrast, our DeCoDe focuses on improving the core diffusion mechanism without any targeted affinity optimization. This difference explains the observed trade-off: AliDiff's gain in binding affinity is achieved by compromising molecular structure quality. For instance, as shown in the table below, AliDiff exhibits consistently higher (i.e., worse) Jensen–Shannon Divergence (JSD) values across key bond lengths compared to its backbone model IPDiff, indicating a degradation in structural fidelity.
> ﻿
> | JSD of Bone Length ↓| IPDiff | AliDiff  | IRDiff | IRDeCo|
> | ----------- | ----------- |----------- |----------- |----------- |
> | C-C | 0.386 | 0.386   | 0.439|0.305|
> | C=C| 0.245  | 0.566   | 0.272|0.251|
> | C-N| 0.298  | 0.410   | 0.302|0.258|
> | C=N| 0.238  | 0.613   | 0.255|0.230|
> | C-O| 0.366  | 0.499   | 0.371|0.303|
> | C=O| 0.353  | 0.432   | 0.361|0.220|
> | C:C| 0.169 | 0.332   | 0.214|0.161|
> | C:N| 0.128  | 0.395   | 0.209|0.117|
>
> More importantly, when the same DPO technique is applied to IRDeCo, the resulting model (IRDeCo+DPO) catches up with and even surpasses AliDiff in binding affinity (see Table 3 in the manuscript). This demonstrates that DeCoDe provides a superior and more robust foundation for subsequent objective-specific optimization, achieving comprehensive improvements without sacrificing structural quality.
>
> ---
>
> **W2. On novelty, and differentiation from AliDiff/IRDiff.**
>
> We thank the reviewer for prompting a clearer articulation of our methodological novelty. The core innovation lies in the decoupling of the diffusion processes​ for global binding position and internal molecular structure—a departure from prior coupled approaches.
>
> Formally, in the global reference frame of the pocket–ligand complex, ligand atom coordinates $X$​ are governed by two factors: the global binding position $X^c$ and the internal structural coordinates $X^s$, i.e., $X=X^c+X^s$.
>
>
> - AliDiff/IRDiff: Directly Diffusing(or Reconstructing) $X$. This is a coupled diffusion process, because $X^c$ and $X^s$ have to share a synchronized diffusion noise $\beta$: $\beta X= \beta (X^c+ X^s)$.
>
> -  Our DeCoDe: Separately Diffusing(or Reconstructing)  $X^c$ and $X^s$. This is a decoupled diffusion process, because we can use asynchronous diffusion noises $\beta^c$ and $\beta^s$ : $\beta X= \beta^c X^c+ \beta^s X^s$.
>
> This is **Diffusion Decoupling**.
>
> Based on this, we can deliberately select such a set of noise schedules $\beta^c$ and $\beta^s$, with which the binding position information  $X^c$ would be restored earlier than the molecular structure information $X^s$ during the reverse process. Because the binding position information is relatively low-frequency, it can be roughly determined earlier and is conducive to the reconstruction of the detailed structure information subsequently.
>
> This is **Time-dependent noise-scheduling**.
>
> This divide-and-conquer approach fundamentally differs from prior SBDD diffusion models, which solve the two subproblems synchronously in a high-dimensional joint space (binding position $ \times $ molecular structure), thereby introducing unnecessary complexity. Our decoupling simplifies the learning problem and enables more precise generation control.
>
> ---
>
> **W3. Figure 6 is not easy to visualize; refer to IRDiff and AliDiff.**
>
> Thank you for this helpful suggestion. We agree that the original visualization could be improved. We have revised Figure 6 by adopting clearer presentation styles similar to those used in IRDiff and AliDiff, as recommended. The updated figure is available for review at: https://anonymous.4open.science/r/anonym056P/figure6.pdf. We will incorporate this improved version into the final manuscript to enhance clarity.
>
> ---
>
> We believe the above responses and additional explanations have fully addressed the reviewer’s concerns. We thank the reviewer again for the constructive feedback, which has been invaluable in strengthening the presentation of our work.

---

> > ### Author Rebuttal · Reviewer_HY4A · 2026-04-03
> >
> > I appreciate the clarifications and I will retain my original score.

---

> > > ### Author Response · Authors · 2026-04-04
> > >
> > > We thank you again for your constructive comments, , which have greatly helped us improve the work. Wishing you a happy life.

---

### Decision · Program_Chairs · 2026-04-30

**Decision:**

Accept (spotlight)

**Comment:**

Reviewers consistently agree that the submission addresses a significant problem in the field. The proposed method is sound and novel, the experimental evaluations are comprehensive and strongly support the claims, and the paper is well written and clearly organized.

Reviewers also raised some minor questions, many of which have been well addressed during the rebuttal period. We suggest that the authors revise the paper accordingly to produce an improved final version.